# Towards Diverse Behaviors: A Benchmark for Imitation Learning with Human Demonstrations

**Xiaogang Jia** [*†‡] **Denis Blessing**[†] **Xinkai Jiang**[†‡] **Moritz Reuss**[‡]
**Atalay Donat**[†] **Rudolf Lioutikov**[‡] **Gerhard Neumann**[†]
[†] Autonomous Learning Robots, Karlsruhe Institute of Technology
[‡] Intuitive Robots Lab, Karlsruhe Institute of Technology

## Abstract

Imitation learning with human data has demonstrated remarkable success in teaching robots in a wide range of skills. However, the inherent diversity in human behavior leads to the emergence of multi-modal data distributions, thereby presenting a formidable challenge for existing imitation learning algorithms. Quantifying a model's capacity to capture and replicate this diversity effectively is still an open problem. In this work, we introduce simulation benchmark environments and the corresponding *Datasets with Diverse human Demonstrations for Imitation Learning (D3IL)*, designed explicitly to evaluate a model's ability to learn multi-modal behavior. Our environments are designed to involve multiple sub-tasks that need to be solved, consider manipulation of multiple objects which increases the diversity of the behavior and can only be solved by policies that rely on closed loop sensory feedback. Other available datasets are missing at least one of these challenging properties. To address the challenge of diversity quantification, we introduce tractable metrics that provide valuable insights into a model's ability to acquire and reproduce diverse behaviors. These metrics offer a practical means to assess the robustness and versatility of imitation learning algorithms. Furthermore, we conduct a thorough evaluation of state-of-the-art methods on the proposed task suite. This evaluation serves as a benchmark for assessing their capability to learn diverse behaviors. Our findings shed light on the effectiveness of these methods in tackling the intricate problem of capturing and generalizing multi-modal human behaviors, offering a valuable reference for the design of future imitation learning algorithms. Project page: https://alrhub.github.io/d3il-website/

## 1 Introduction

Imitation Learning (IL) (Osa et al., 2018) from human expert data has emerged as a powerful approach for imparting a wide array of skills to robots and autonomous agents. In this setup, a human expert controls the robot, e.g., using tele-operation interfaces, and demonstrates solutions to complex tasks. Leveraging human expertise, IL algorithms have demonstrated remarkable success in training robots to perform a wide range of complex tasks with finesse (Brohan et al., 2022; Zhao et al., 2023; Huang et al., 2023). However, learning from human data is challenging due to the inherent diversity in human behavior (Grauman et al., 2022; Lynch et al., 2020). This diversity, arising from factors such as individual preferences, noise, varying levels of expertise, and different problem-solving approaches, poses a formidable hurdle for existing imitation learning algorithms.

In recent years, there has been a notable surge of interest in the development of methods aimed at capturing diverse behaviors (Shafiullah et al., 2022; Chi et al., 2023; Reuss et al., 2023). These endeavors are driven by various objectives, including improving generalization capabilities (Merel et al., 2018; Li et al., 2017), enhancing skill transfer between policies (Merel et al., 2018; Kumar et al., 2020), and gaining advantages in competitive games (Celik et al., 2022), among others. Nevertheless, these approaches are often tested on synthetically generated datasets, datasets with limited

---

[*]Correspondence to xiaogang.jia@kit.edu

|  | D4RL | Robomimic | T-Shape | Relay Kitchen | Block-Push | D3IL (ours) |
|---|---|---|---|---|---|---|
| Human Demonstrations | ✓ | ✓ | ✓ | ✓ | ✗ | ✓ |
| State Data | ✓ | ✓ | ✓ | ✓ | ✓ | ✓ |
| Visual Data | ✗ | ✓ | ✓ | ✗ | ✓ | ✓ |
| Diverse Behavior | ✗ | ✗ | ✗ | ✓ | ✓ | ✓ |
| Quant. Diverse Behavior | ✗ | ✗ | ✗ | ✗ | ✓ | ✓ |

Table 1: Comparison between existing imitation learning benchmarks: D4RL (Fu et al., 2020), Robomimic (Mandlekar et al., 2021) are used to benchmark offline RL algorithms, but do not focus on diversity. T-Shape Pushing (Florence et al., 2022; Chi et al., 2023) has 2 solutions and is a relatively simple task, while Relay Kitchen (Gupta et al., 2019) has a higher diversity in terms of tasks, however, the behaviors do not require closed-loop feedback due to the limited task variability. Block-Push shows diverse behavior (Florence et al., 2022), however, it is generated by a scripted policy, resulting in a reduced variance and missing fallback strategies.

diverse behavior or limited need for closed-loop feedback. Other datasets are collected directly on real robot platforms and lack simulation environments that can be used for benchmarking other algorithms. While real robot environments are of course preferable to show the applicability of the approaches on the real system, they hinder benchmarking new algorithms against the reported results as that would require rebuilding the real robot setup, which is often infeasible. Furthermore, the majority of existing studies do not provide metrics to quantitatively measure a model's ability to replicate diverse behaviors, often relying solely on qualitative analysis (Shafiullah et al., 2022).

To address these challenges, this work introduces benchmark environments and the corresponding *Datasets with Diverse Human Demonstrations for Imitation Learning (D3IL)*. Our primary aim is to provide a comprehensive evaluation framework that explicitly assesses an algorithm's ability to learn from multi-modal data distributions. We have designed these datasets to encompass the richness and variability inherent in human behavior, offering more realistic and challenging benchmark scenarios for imitation learning. Moreover, the given environments are challenging as the behavior is composed of multiple sub-tasks and they require policies that heavily rely on closed-loop feedback. To tackle the intricate problem of quantifying a model's capacity to capture and replicate diverse human behaviors, we introduce tractable metrics. These metrics provide valuable insights into a model's versatility and adaptability, allowing for a more nuanced assessment of imitation learning algorithms' performance.

Furthermore, we conduct a rigorous evaluation of state-of-the-art imitation learning methods using the D3IL task suite. This evaluation serves as a benchmark for assessing the capability of these methods to learn diverse behaviors, shedding light on their effectiveness, hyper-parameter choices, and the representations they employ by. By analyzing their performance for these realistic human-generated datasets in challenging environments. Here, our analysis includes different backbones of the different IL architectures (MLPs and various versions of transformers), different ways to capture the multi-modality of the action distribution (clustering, VAEs, IBC, various versions of diffusion). Moreover, we analyze the performance using the direct state observations or image observations as input to the policies and draw conclusions on the benefits and drawbacks of these representations. Finally, we evaluate which method can deal best with small datasets. Using all these insights, our study will inform the design and advancement of future imitation learning algorithms.

## 2 RELATED WORK

This section provides an overview of existing benchmarks and related research in the field of robot learning and imitation learning. Table 1 presents a comprehensive overview of the distinctive features that set our work apart from the most closely related benchmarks in the field.

RLBench (James et al., 2020) and ManiSkill2 (Mu et al., 2021; Gu et al., 2023) are two recent large-scale benchmarks for robot learning with a large variety of tasks with both, proprioceptive and visual data. However, the demonstrations are generated through motion planning and are hence lacking diversity. Meta-World is another robot learning benchmark with 50 different tasks and a focus on multi-task reinforcement learning (Yu et al., 2020). Yet, no human demonstrations exists for these tasks and most tasks typically only have one solution. D4RL (Fu et al., 2020) proposed

standardized environments and datasets for offline reinforcement learning with various tasks. Several other benchmarks specialize in areas such as language-guided multitask learning in simulation (Mees et al., 2022; Lynch et al., 2023; Gong et al., 2023; Zeng et al., 2021; Jiang et al., 2023; Gong et al., 2023), continual learning (Liu et al., 2023) and diverse multi-task real world skill learning (Walke et al., 2023; Bharadhwaj et al., 2023; Heo et al., 2023). A benchmark with a focus on human demonstrations is Robomimic (Mandlekar et al., 2021), featuring 8 different task in simulation and real world environments. The demonstrations, collected by multiple people with various degrees of expertise using the RoboTurk framework (Mandlekar et al., 2018). Yet, the tasks where not designed to show diverse behavior and consequently, non of the tested algorithms was designed to capture the diversity of the behavior.

The benchmarks most closely related to D3IL are Block-Push (Florence et al., 2022; Shafiullah et al., 2022), T-Shape Pushing (Florence et al., 2022) and Relay Kitchen Gupta et al. (2019). While Block-Push has 4 different modalities, the behavior is generated by a scripted policy and is hence lacking variance and fall-back solutions in the dataset. T-Shape tasks are limited to a maximum of two modalities. The Relay Kitchen environment (Gupta et al., 2019) contains 7 different tasks with human demonstrations that solve four tasks in sequence but does not require closed-loop sensory feedback. Further, several recent work have shown to achieve nearly optimal performance (Chi et al., 2023; Reuss et al., 2023) on these benchmarks, limiting further progress in these environments. Moreover, none of the mentioned benchmarks offer metrics to evaluate the diversity of the learned behavior.

## 3 DATASETS WITH DIVERSE DEMONSTRATIONS FOR IMITATION LEARNING

In this section, we present a novel suite of tasks known as D3IL - Datasets with Diverse Demonstrations for Imitation Learning. We first introduce metrics for the quantification of diverse behavior (refer to Section 3.1). Subsequently, we delve into the design principles underlying our tasks, which are subsequently introduced at the end of this section.

### 3.1 QUANTIFYING DIVERSE BEHAVIOR

Let $\mathcal{D} = \{(\mathbf{a}_n, \mathbf{s}_n)\}_{n=1}^N$ be a human-recorded demonstration dataset with actions $\mathbf{a} \in \mathcal{A}$ and states $\mathbf{s} \in \mathcal{S}$. Further, let $p(\mathbf{a}|\mathbf{s})$ be the underlying state-conditional action distribution. The goal of imitation learning is to learn a policy $\pi(\mathbf{a}|\mathbf{s}) \approx p(\mathbf{a}|\mathbf{s})$. Diversity is characterized by a multimodal action distribution $p(\mathbf{a}|\mathbf{s})$, meaning that there are multiple distinct actions that are likely for a given state $\mathbf{s}$. We refer to this as multimodality on a *state-level*. However, this multimodality is hard to quantify in most scenarios as we do not have access to $p(\mathbf{a}|\mathbf{s})$. Instead, we look at the multimodality on a *behavior-level* to define an auxiliary metric that reflects if a model is capable of imitating diverse behaviors. To define behavior-level multi-modality, we introduce discrete behavior descriptors $\beta \in \mathcal{B}$, for example, which box has been chosen to be pushed first. The space $\mathcal{B}$ of behavior descriptors are thus task-specific and are discussed in Section 3.3.

To evaluate the ability of a trained policy $\pi(\mathbf{a}|\mathbf{s})$ to capture multi-modality, we collected data such that we have an approximately equal number of demonstrations for each behavior descriptor $\beta \in \mathcal{B}$. Thereafter, we perform simulations in the task environment to compute the agents' distribution $\pi(\beta)$ over its achieved behaviors. We then assess the policy's capability of learning diverse behavior by leveraging the *behavior entropy*, that is,

$$\mathcal{H}\big(\pi(\beta)\big) = -\sum_{\beta \in \mathcal{B}} \pi(\beta) \log_{|\mathcal{B}|} \pi(\beta). \tag{1}$$

 Please note that we employ the logarithm with a base of $|\mathcal{B}|$ to ensure that $\mathcal{H}\big(\pi(\beta)\big) \in [0,1]$. This choice of base facilitates a straightforward interpretation of the metric: An entropy value of $0$ signifies a policy that consistently executes the same behavior, while an entropy value of $1$ represents a diverse policy that executes all behaviors with equal probability, that is, $\pi(\beta) \approx 1/|\mathcal{B}|$ and hence matches the true behavior distribution by the design of the data collection process. Yet, a high behavior entropy can also be achieved by following different behaviors in different initial states in a deterministic manner, and hence, this behavior entropy can be a poor metric for behavior diversity. Hence, we evaluate the expected entropy of the behavior conditioned on the initial state $\mathbf{s}_0$. If, for the same initial state, all behaviors can be achieved, the conditional behavior entropy is high, while

if for the same initial state always the same behavior is executed, this entropy is 0. We define the conditional behavior entropy as

$$\mathbb{E}_{\mathbf{s}_0 \sim p(\mathbf{s}_0)} \left[ \mathcal{H}\big(\pi(\beta|\mathbf{s}_0)\big) \right] \approx -\frac{1}{S_0} \sum_{\mathbf{s}_0 \sim p(\mathbf{s}_0)} \sum_{\beta \in \mathcal{B}} \pi(\beta|\mathbf{s}_0) \log_{|\mathcal{B}|} \pi(\beta|\mathbf{s}_0), \qquad (2)$$

where the expectation is approximated using a Monte Carlo estimate. Here, $S_0$ denotes the number of samples from the initial state distribution $p(\mathbf{s}_0)$.

## 3.2 TASK DESIGN PRINCIPLES

We build D3IL based on the following key principles: **i) Diverse Behavior.** Diversity is the central aspect of our task design. We intentionally design our tasks to encompass multiple viable approaches to successful task completion. To quantify the behavior diversity, we explicitly specify these distinct behaviors, each representing a legitimate solution. **ii.) Multiple Human Demonstrators.** To reflect the natural variability in human behavior and to obtain a richer dataset, we have collected demonstration data from multiple human demonstrators. This diversity in data sources introduces variations in the quality and style of demonstrations. **iii.) Variable Trajectory Lengths.** Our tasks incorporate variable trajectory lengths, replicating real-world scenarios where demonstrations may differ in duration. This design choice challenges our learning agents to handle non-uniform data sequences effectively. By accommodating varying trajectory lengths, our approach must learn to adapt and generalize to different time horizons, a critical property for real-world applications. **iv.) Task Variations and Closed-Loop Feedback.** For most tasks, agents need to rely on sensory feedback to achieve a good performance which considerably increases the complexity of the learning task in comparison to learning open-loop trajectories. We achieve this by introducing task variations in every execution. For example, in every execution, the initial position of the objects will be different and the agent needs to adapt its behaviour accordingly.

## 3.3 TASK DESCRIPTION AND DATA COLLECTION

We simulate all tasks (**T1**-**T5**) using the Mujoco physics engine (Todorov et al. (2012)). The simulation environment consists of a 7DoF Franka Emika Panda robot and various objects. For **T1**-**T4**, the robot is equipped with a cylindrical end effector to perform object manipulations. The corresponding demonstrations are recorded by using an Xbox game-pad which sends commands to an inverse kinematics (IK) controller. In **T5**, the robot has a parallel gripper and uses an augmented reality setup for controlling the end effector (Jiang et al., 2024), allowing for more dexterous manipulations.

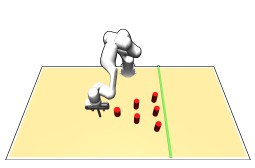

**Avoiding (T1).** The Avoiding task requires the robot to reach the green finish line from a fixed initial position without colliding with one of the six obstacles. The task does not require object manipulation and is designed to test a model's ability to capture diversity. There are 24 different ways of successfully completing the task and thus $|\mathcal{B}| = 24$. The dataset contains 96 demonstrations in total, comprising 24 solutions with 4 trajectories for each solution.

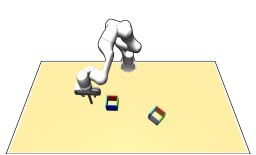

**Aligning (T2).** The Aligning task requires the robot to push a hollow box to a predefined target position and orientation. The task can be completed by either pushing the box from outside or inside and thus $|\mathcal{B}| = 2$. It requires less diversity than T1 but involves complex object manipulation. The dataset contains 1k demonstrations, 500 for each behavior with uniformly sampled initial states.

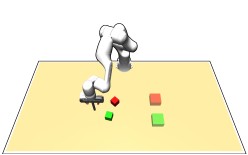

**Pushing (T3).** This task requires the robot to push two blocks to fixed target zones. Having two blocks and two target zones results in $|\mathcal{B}| = 4$ behaviors. The task has more diversity than T2 and involves complex object manipulations. Additionally, the task has high variation in trajectory length caused by multiple human demonstrators with different experience levels. The dataset contains 2k demonstrations, 500 for each behavior with uniformly sampled initial block positions.

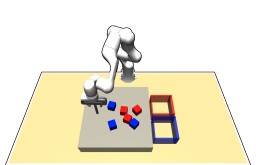

**Sorting-X (T4).** This task requires the robot to sort red and blue blocks to their color-matching target box. The 'X' refers to the number of blocks. In this work, we test three difficulty levels, i.e., $X \in \{2, 4, 6\}$. The number of behaviors $|\mathcal{B}|$ is determined by the sorting order. For $X = 6$, the task has many objects, is highly diverse ($|\mathcal{B}| = 20$), requires complex manipulations, has high variation in trajectory length, and is thus more challenging than T1-T3. The dataset contains 1.6k demonstrations for $X = 6$ and an approximately equal number of trajectories for each sorting order.

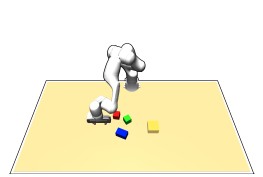

**Stacking-X (T5).** This task requires the robot to stack blocks with different colors in a (yellow) target zone. The number of behaviors $|\mathcal{B}|$ is given by the stacking order. Having three different blocks results in six different combinations for stacking the blocks and hence $|\mathcal{B}| = 6$. This task additionally requires dexterous manipulations since the blue box has the be stacked upright and is thus considered to be the most challenging out of all five tasks. Here, 'X' does not refer to the difficulty but rather to the evaluation protocol: For Stacking-X with $X \in \{1, 2, 3\}$ we compute the performance metrics of the models that are capable of stacking X blocks. The $\approx$ 1k trajectories were recorded using augmented reality (AR) such that each stacking order is equally often present.

## 4 BENCHMARKING ON D3IL TASKS

### 4.1 IMITATION LEARNING ALGORITHMS

We benchmark a large set of recent imitation learning algorithms. These algorithms can be categorized along three axes. The first axis specifies whether the algorithms are using a history of state observations. For algorithms that do not use history (Sohn et al., 2015; Florence et al., 2022), a standard MLP is used as backbone architecture, while for history-dependent policies (Shafiullah et al., 2022; Pearce et al., 2023; Reuss et al., 2023; Chi et al., 2023) we use a causal masked Transformer Decoder (GPT) based on the implementation of BeT (Shafiullah et al., 2022) as backbone as this architecture has been also adopted by other recent approaches. In our experiments, we use a history of $k = 5$. The second axis categorizes whether the algorithms are predicting only a single action or the future action sequence (Zhao et al., 2023; Chi et al., 2023), which has been recently introduced as

| Methods | History | Backbone | Prediction | Diversity |
|---|---|---|---|---|
| BC-MLP | $k=1$ | MLP | SA | Deterministic |
| VAE-State (Sohn et al., 2015) | $k=1$ | MLP | SA | VAE |
| BeT-MLP | $k=1$ | MLP | SA | Cluster |
| IBC (Florence et al., 2022) | $k=1$ | MLP | SA | Implicit |
| DDPM-MLP | $k=1$ | MLP | SA | Disc. Diffusion |
| VAE-ACT (Zhao et al., 2023) | $k=1$ | T-EncDec | AC | VAE |
| BC-GPT | $k=5$ | GPT | SA | Deterministic |
| BeT (Shafiullah et al., 2022) | $k=5$ | GPT | SA | Cluster |
| DDPM-GPT (Pearce et al., 2023) | $k=5$ | GPT | SA | Disc. Diffusion |
| BESO (Reuss et al., 2023) | $k=5$ | GPT | SA | Cont. Diffusion |
| DDPM-ACT (Chi et al., 2023) | $k=5$ | T-EncDec | AC | Disc. Diffusion |

Table 2: Categorization of the tested IL algorithms. The algorithms differ in whether they use history information (MLP backbones for no history, Transformer/GPT backbones for history-based models), whether they predict future action sequences (single actions/SA or action chunking/AC, Transformer-Encoder-Decoder/T-EncDec backbones for AC) and how they model diverse behavior (last column).

action chunking which has been shown to improve performance in some tasks. For action chunking models, we use a Transformer Encoder Decoder structure (T-EncDec) (Vaswani et al., 2017) based on the implementation of VAE-ACT (Zhao et al., 2023). The third axis categorizes how the algorithms capture behavior diversity. Standard deterministic models are not able to do so, while more recent models use clustering (Shafiullah et al., 2022), VAEs (Zhao et al., 2023; Sohn et al., 2015), implicit models (Florence et al., 2022) as well as discrete-time (Pearce et al., 2023; Chi et al., 2023), or continuous time diffusion models (Reuss et al., 2023). To improve our understanding of the influence of the specific design choices, we also evaluate new variants of these combinations such as history-free (MLP-based) action clustering (BeT-MLP). The full table of used algorithmic setups is depicted in Table 2. A more detailed description of all model architectures is given in Appendix C.

## 4.2 EXPERIMENTAL SETUP

We briefly outline the most important aspects of our experimental setup.

**Evaluation Protocol.** To assess a model's ability to capture diverse behavior, we propose the *behavior entropy* and the *conditional behavior entropy* (Section 3.1). For the former, we perform multiple simulations, from which we compute the model's behavior distribution $\pi(\beta)$. For the latter, we use a randomly sampled but fixed set of initial states $s_0$ and perform multiple simulations for each $s_0$ in order to compute the conditional behavior distribution $\pi(\beta|s_0)$. Alongside the entropy, we also report the *success rate* which is the fraction of simulations that led to successful task completion.

**State / Observation Representation.** For most experiments, we provide state representation and image observations. For the former, we use handcrafted features that work well empirically. For the latter, we used two different camera views, in-hand and front view with a $96 \times 96$ image resolution. We follow Chi et al. (2023) and use a ResNet-18 architecture as an image encoder for all methods.

**Model Selection.** Model selection is challenging as the training objective does not coincide with task performance (Gulcehre et al., 2020; Paine et al., 2020; Fu et al., 2021; Mandlekar et al., 2021). Moreover, policy performance can exhibit substantial variations from one training epoch to another (Mandlekar et al., 2021). For image-based experiments we evaluate the task performance frequently (after every $1/10$th of total training) and chose the model with the best task-performance. Due to high-computational demand for simulation, we split the dataset into training (90%) and validation (10%) and chose the model with the lowest validation loss for state-based experiments. We ensure that the training converges by using 500 epochs for state representation and 200 for image observations. We extensively tune the most important hyperparameters of all methods using Bayesian optimization (Snoek et al., 2012). We report the mean and the standard deviation over six random seeds for all experiments.

## 4.3 BENCHMARK USING STATE REPRESENTATION

We present the results utilizing state representations in Table 3. Detailed information regarding individual state representations for the tasks can be found in Appendix A. Through an analysis of the performance and diversity across all methods, we outline our key findings below.

**Multiple Strategies for Multi-Modal Behavior Learning.** This section discusses various strategies employed by different models for learning multi-modal behavior. Deterministic models like BC-MLP and BC-GPT have limitations in this regard since they can only generate a single solution given an initial state $s_0$. On the other hand, VAE-based models like VAE-State and VAE-ACT are capable of generating different behaviors, but they tend to exhibit low behavior entropy, presumably due to a phenomenon often referred to as "mode collapse" (Kingma & Welling, 2013). In contrast, IBC, BeT, and diffusion-based models demonstrate the capacity to learn multi-modal distributions. IBC, for instance, exhibits high diversity in tasks like Avoiding, Pushing, and Sorting-2, with entropy levels slightly superior to DDPM-MLP. BeT is capable of learning diverse solutions for each task, but it comes with a significant performance drop when compared to the deterministic BC-GPT baseline. Notably, diffusion-based methods, especially those incorporating transformer backbones, excel in learning diverse behavior while maintaining strong performance across all tasks. To offer additional insights, we include visual representations of the diverse solutions generated by each method for the Avoiding task in Figure 10. Furthermore, we conduct an in-depth analysis of hyperparameter sensitivity, which is discussed in Section B. This analysis provides a deeper understanding

Table 3: Comparison between various Imitation Learning algorithms, some of which incorporate history (‡), action chunking (†), or both using state representations (State Data) and image observations (Visual Data). We present the mean and standard deviation across six random seeds, highlighting the best performance for both history-based and non-history-based models using bold formatting.

| | Success Rate | Entropy | Success Rate | Entropy | Success Rate | Entropy |
|---|---|---|---|---|---|---|
| **State Data** | Avoiding (T1) | | Aligning (T2) | | Pushing (T3) | |
| BC-MLP | $0.666_{\pm0.512}$ | $0.0_{\pm0.0}$ | $0.708_{\pm0.052}$ | $0.0_{\pm0.0}$ | $0.522_{\pm0.165}$ | $0.0_{\pm0.0}$ |
| VAE-State | $0.716_{\pm0.265}$ | $0.195_{\pm0.111}$ | $0.579_{\pm0.138}$ | $0.030_{\pm0.053}$ | $\mathbf{0.604_{\pm0.087}}$ | $0.170_{\pm0.031}$ |
| BeT-MLP | $0.665_{\pm0.136}$ | $0.836_{\pm0.071}$ | $0.507_{\pm0.081}$ | $0.485_{\pm0.092}$ | $0.420_{\pm0.049}$ | $0.628_{\pm0.123}$ |
| IBC | $\mathbf{0.760_{\pm0.046}}$ | $\mathbf{0.850_{\pm0.038}}$ | $0.638_{\pm0.027}$ | $0.300_{\pm0.048}$ | $0.574_{\pm0.048}$ | $\mathbf{0.816_{\pm0.028}}$ |
| DDPM-MLP | $0.637_{\pm0.055}$ | $0.801_{\pm0.034}$ | $\mathbf{0.763_{\pm0.039}}$ | $\mathbf{0.712_{\pm0.064}}$ | $0.569_{\pm0.047}$ | $0.796_{\pm0.043}$ |
| VAE-ACT[†] | $0.851_{\pm0.109}$ | $0.224_{\pm0.173}$ | $\mathbf{0.891_{\pm0.022}}$ | $0.025_{\pm0.012}$ | $\mathbf{0.951_{\pm0.033}}$ | $0.070_{\pm0.029}$ |
| BC-GPT[‡] | $0.833_{\pm0.408}$ | $0.0_{\pm0.0}$ | $0.833_{\pm0.043}$ | $0.0_{\pm0.0}$ | $0.855_{\pm0.054}$ | $0.0_{\pm0.0}$ |
| BeT[‡] | $0.747_{\pm0.030}$ | $0.844_{\pm0.035}$ | $0.645_{\pm0.069}$ | $0.536_{\pm0.105}$ | $0.724_{\pm0.051}$ | $0.788_{\pm0.031}$ |
| DDPM-GPT[‡] | $0.927_{\pm0.022}$ | $\mathbf{0.898_{\pm0.018}}$ | $0.839_{\pm0.020}$ | $0.664_{\pm0.075}$ | $0.847_{\pm0.050}$ | $0.862_{\pm0.017}$ |
| BESO[‡] | $\mathbf{0.950_{\pm0.015}}$ | $0.856_{\pm0.018}$ | $0.861_{\pm0.016}$ | $0.711_{\pm0.030}$ | $0.794_{\pm0.095}$ | $\mathbf{0.871_{\pm0.037}}$ |
| DDPM-ACT[†,‡] | $0.899_{\pm0.006}$ | $0.863_{\pm0.068}$ | $0.849_{\pm0.023}$ | $\mathbf{0.749_{\pm0.041}}$ | $0.920_{\pm0.027}$ | $0.859_{\pm0.012}$ |
| **State Data** | Sorting-2 (T4) | | Sorting-4 (T4) | | Sorting-6 (T4) | |
| BC-MLP | $0.444_{\pm0.069}$ | $0.0_{\pm0.0}$ | $0.058_{\pm0.048}$ | $0.0_{\pm0.0}$ | $0.016_{\pm0.014}$ | $0.0_{\pm0.0}$ |
| VAE-State | $0.451_{\pm0.059}$ | $0.106_{\pm0.051}$ | $0.079_{\pm0.042}$ | $0.090_{\pm0.032}$ | $\mathbf{0.030_{\pm0.016}}$ | $\mathbf{0.086_{\pm0.017}}$ |
| BeT-MLP | $0.317_{\pm0.070}$ | $0.309_{\pm0.059}$ | $0.003_{\pm0.001}$ | $0.021_{\pm0.052}$ | $0.0_{\pm0.0}$ | $0.0_{\pm0.0}$ |
| IBC | $0.459_{\pm0.033}$ | $\mathbf{0.370_{\pm0.071}}$ | $\mathbf{0.080_{\pm0.016}}$ | $\mathbf{0.097_{\pm0.026}}$ | $0.007_{\pm0.005}$ | $0.024_{\pm0.028}$ |
| DDPM-MLP | $\mathbf{0.460_{\pm0.039}}$ | $0.327_{\pm0.037}$ | $0.032_{\pm0.009}$ | $0.071_{\pm0.025}$ | $0.0_{\pm0.0}$ | $0.0_{\pm0.0}$ |
| VAE-ACT[†] | $0.858_{\pm0.073}$ | $0.012_{\pm0.018}$ | $\mathbf{0.565_{\pm0.108}}$ | $0.114_{\pm0.020}$ | $0.021_{\pm0.019}$ | $0.057_{\pm0.040}$ |
| BC-GPT[‡] | $0.800_{\pm0.040}$ | $0.0_{\pm0.0}$ | $0.075_{\pm0.063}$ | $0.0_{\pm0.0}$ | $0.007_{\pm0.003}$ | $0.0_{\pm0.0}$ |
| BeT[‡] | $0.680_{\pm0.060}$ | $\mathbf{0.473_{\pm0.049}}$ | $0.018_{\pm0.010}$ | $0.059_{\pm0.035}$ | $0.0_{\pm0.0}$ | $0.0_{\pm0.0}$ |
| DDPM-GPT[‡] | $0.854_{\pm0.013}$ | $0.431_{\pm0.060}$ | $0.337_{\pm0.084}$ | $\mathbf{0.372_{\pm0.068}}$ | $0.005_{\pm0.003}$ | $0.031_{\pm0.029}$ |
| BESO[‡] | $0.784_{\pm0.026}$ | $0.387_{\pm0.038}$ | $0.188_{\pm0.042}$ | $0.232_{\pm0.040}$ | $\mathbf{0.020_{\pm0.002}}$ | $\mathbf{0.055_{\pm0.031}}$ |
| DDPM-ACT[†,‡] | $\mathbf{0.882_{\pm0.023}}$ | $0.347_{\pm0.048}$ | $0.300_{\pm0.060}$ | $0.300_{\pm0.051}$ | $0.0_{\pm0.0}$ | $0.0_{\pm0.0}$ |
| **State Data** | Stacking-1 (T5) | | Stacking-2 (T5) | | Stacking-3 (T5) | |
| BC-MLP | $0.030_{\pm0.022}$ | $0.0_{\pm0.0}$ | $0.0_{\pm0.0}$ | $0.0_{\pm0.0}$ | $0.0_{\pm0.0}$ | $0.0_{\pm0.0}$ |
| VAE-State | $0.036_{\pm0.023}$ | $0.064_{\pm0.092}$ | $0.001_{\pm0.001}$ | $0.0_{\pm0.0}$ | $0.0_{\pm0.0}$ | $0.0_{\pm0.0}$ |
| BeT-MLP | $0.102_{\pm0.059}$ | $\mathbf{0.120_{\pm0.031}}$ | $0.003_{\pm0.001}$ | $0.0_{\pm0.0}$ | $0.0_{\pm0.0}$ | $0.0_{\pm0.0}$ |
| IBC | $0.010_{\pm0.016}$ | $0.013_{\pm0.027}$ | $0.0_{\pm0.0}$ | $0.0_{\pm0.0}$ | $0.0_{\pm0.0}$ | $0.0_{\pm0.0}$ |
| DDPM-MLP | $\mathbf{0.244_{\pm0.161}}$ | $0.113_{\pm0.084}$ | $\mathbf{0.042_{\pm0.045}}$ | $\mathbf{0.011_{\pm0.014}}$ | $\mathbf{0.001_{\pm0.0009}}$ | $0.096_{\pm0.193}$ |
| VAE-ACT[†] | $0.308_{\pm0.096}$ | $0.095_{\pm0.027}$ | $0.083_{\pm0.067}$ | $0.037_{\pm0.037}$ | $0.0_{\pm0.0}$ | $0.0_{\pm0.0}$ |
| BC-GPT[‡] | $0.241_{\pm0.055}$ | $0.0_{\pm0.0}$ | $0.025_{\pm0.016}$ | $0.0_{\pm0.0}$ | $0.0_{\pm0.0}$ | $0.0_{\pm0.0}$ |
| BeT[‡] | $0.163_{\pm0.066}$ | $0.234_{\pm0.079}$ | $0.025_{\pm0.015}$ | $0.051_{\pm0.035}$ | $0.001_{\pm0.001}$ | $0.0_{\pm0.0}$ |
| DDPM-GPT[‡] | $0.895_{\pm0.013}$ | $0.222_{\pm0.052}$ | $\mathbf{0.708_{\pm0.031}}$ | $0.146_{\pm0.029}$ | $\mathbf{0.120_{\pm0.018}}$ | $0.120_{\pm0.039}$ |
| BESO[‡] | $\mathbf{0.909_{\pm0.007}}$ | $0.382_{\pm0.053}$ | $0.685_{\pm0.034}$ | $\mathbf{0.177_{\pm0.058}}$ | $0.095_{\pm0.006}$ | $0.131_{\pm0.028}$ |
| DDPM-ACT[†,‡] | $0.634_{\pm0.049}$ | $\mathbf{0.762_{\pm0.048}}$ | $0.300_{\pm0.053}$ | $0.503_{\pm0.070}$ | $0.096_{\pm0.021}$ | $\mathbf{0.190_{\pm0.057}}$ |
| **Image Data** | Aligning (T2) | | Sorting-4 (T4) | | Sorting-6 (T4) | |
| BC-MLP | $0.148_{\pm0.045}$ | $0.0_{\pm0.0}$ | $0.666_{\pm0.080}$ | $0.0_{\pm0.0}$ | $0.636_{\pm0.026}$ | $0.0_{\pm0.0}$ |
| VAE-State | $0.121_{\pm0.035}$ | $0.013_{\pm0.019}$ | $0.653_{\pm0.061}$ | $0.165_{\pm0.010}$ | $0.570_{\pm0.023}$ | $0.236_{\pm0.013}$ |
| BeT-MLP | $0.203_{\pm0.035}$ | $0.053_{\pm0.028}$ | $0.671_{\pm0.014}$ | $\mathbf{0.296_{\pm0.033}}$ | $0.616_{\pm0.035}$ | $\mathbf{0.365_{\pm0.022}}$ |
| IBC | $0.179_{\pm0.028}$ | $\mathbf{0.071_{\pm0.052}}$ | $0.707_{\pm0.031}$ | $0.236_{\pm0.027}$ | $0.664_{\pm0.039}$ | $0.328_{\pm0.026}$ |
| DDPM-MLP | $\mathbf{0.233_{\pm0.029}}$ | $0.060_{\pm0.033}$ | $\mathbf{0.763_{\pm0.028}}$ | $0.237_{\pm0.028}$ | $\mathbf{0.748_{\pm0.026}}$ | $0.359_{\pm0.025}$ |
| VAE-ACT[†] | $0.166_{\pm0.017}$ | $0.021_{\pm0.011}$ | $0.717_{\pm0.038}$ | $0.217_{\pm0.024}$ | $0.604_{\pm0.017}$ | $0.276_{\pm0.007}$ |
| BC-GPT[‡] | $0.084_{\pm0.015}$ | $0.0_{\pm0.0}$ | $0.654_{\pm0.048}$ | $0.0_{\pm0.0}$ | $0.590_{\pm0.017}$ | $0.0_{\pm0.0}$ |
| BeT[‡] | $0.154_{\pm0.067}$ | $0.023_{\pm0.012}$ | $0.676_{\pm0.035}$ | $0.305_{\pm0.029}$ | $0.617_{\pm0.033}$ | $0.380_{\pm0.019}$ |
| DDPM-GPT[‡] | $0.316_{\pm0.026}$ | $0.089_{\pm0.021}$ | $\mathbf{0.770_{\pm0.022}}$ | $0.270_{\pm0.033}$ | $0.710_{\pm0.029}$ | $0.314_{\pm0.042}$ |
| BESO[‡] | $\mathbf{0.359_{\pm0.031}}$ | $\mathbf{0.151_{\pm0.051}}$ | $0.764_{\pm0.026}$ | $0.281_{\pm0.035}$ | $\mathbf{0.719_{\pm0.025}}$ | $0.352_{\pm0.018}$ |
| DDPM-ACT[†,‡] | $0.278_{\pm0.071}$ | $0.139_{\pm0.054}$ | $0.659_{\pm0.063}$ | $\mathbf{0.346_{\pm0.028}}$ | $0.643_{\pm0.036}$ | $\mathbf{0.397_{\pm0.025}}$ |
| **Image Data** | Stacking-1 (T5) | | Stacking-2 (T5) | | Stacking-3 (T5) | |
| BC-MLP | $0.531_{\pm0.054}$ | $0.0_{\pm0.0}$ | $\mathbf{0.133_{\pm0.059}}$ | $0.0_{\pm0.0}$ | $0.0_{\pm0.0}$ | $0.0_{\pm0.0}$ |
| VAE-State | $0.517_{\pm0.059}$ | $0.038_{\pm0.022}$ | $0.089_{\pm0.050}$ | $\mathbf{0.028_{\pm0.032}}$ | $0.001_{\pm0.001}$ | $0.0_{\pm0.0}$ |
| BeT-MLP | $0.467_{\pm0.046}$ | $\mathbf{0.195_{\pm0.044}}$ | $0.068_{\pm0.024}$ | $0.027_{\pm0.015}$ | $0.0_{\pm0.0}$ | $0.0_{\pm0.0}$ |
| IBC | $0.002_{\pm0.003}$ | $0.0_{\pm0.0}$ | $0.0_{\pm0.0}$ | $0.0_{\pm0.0}$ | $0.0_{\pm0.0}$ | $0.0_{\pm0.0}$ |
| DDPM-MLP | $\mathbf{0.645_{\pm0.050}}$ | $0.137_{\pm0.061}$ | $0.111_{\pm0.049}$ | $0.027_{\pm0.019}$ | $0.002_{\pm0.002}$ | $0.0_{\pm0.0}$ |
| VAE-ACT[†] | $0.572_{\pm0.081}$ | $0.219_{\pm0.051}$ | $0.196_{\pm0.057}$ | $0.115_{\pm0.037}$ | $0.018_{\pm0.010}$ | $0.006_{\pm0.009}$ |
| BC-GPT[‡] | $0.515_{\pm0.092}$ | $0.0_{\pm0.0}$ | $0.143_{\pm0.037}$ | $0.0_{\pm0.0}$ | $0.003_{\pm0.002}$ | $0.0_{\pm0.0}$ |
| BeT[‡] | $0.500_{\pm0.054}$ | $0.325_{\pm0.082}$ | $0.143_{\pm0.031}$ | $0.057_{\pm0.022}$ | $0.002_{\pm0.001}$ | $0.0_{\pm0.0}$ |
| DDPM-GPT[‡] | $0.704_{\pm0.024}$ | $\mathbf{0.349_{\pm0.053}}$ | $0.153_{\pm0.035}$ | $0.145_{\pm0.050}$ | $0.007_{\pm0.004}$ | $0.012_{\pm0.019}$ |
| BESO[‡] | $0.655_{\pm0.061}$ | $0.182_{\pm0.020}$ | $0.188_{\pm0.041}$ | $0.087_{\pm0.030}$ | $0.003_{\pm0.003}$ | $0.008_{\pm0.020}$ |
| DDPM-ACT[†,‡] | $\mathbf{0.752_{\pm0.045}}$ | $0.334_{\pm0.073}$ | $\mathbf{0.513_{\pm0.058}}$ | $\mathbf{0.302_{\pm0.064}}$ | $\mathbf{0.244_{\pm0.043}}$ | $\mathbf{0.093_{\pm0.042}}$ |

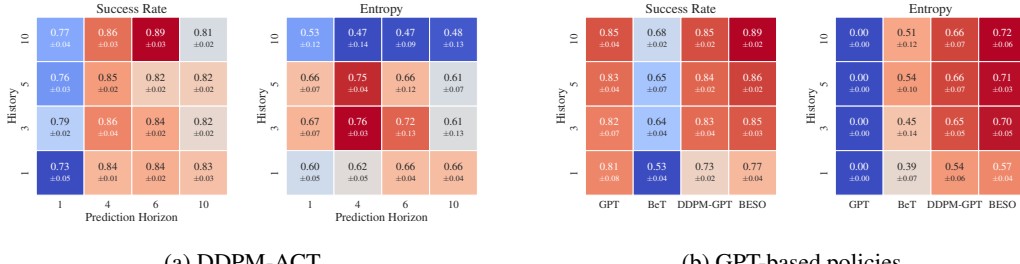

Figure 1: Ablation study for different history lengths and prediction horizons on the Aligning task.

of how hyperparameters influence the ability to learn diverse solutions. Notably, our findings reveal that increasing the number of diffusion steps leads to a substantial enhancement in diversity while maintaining consistent task performance.

**Historical Inputs and Prediction Horizons are Important.** A comparison of results between BC-MLP and BC-GPT, VAE-State and VAE-ACT, BeT-MLP and BeT, DDPM-MLP and DDPM-GPT reveals a notable trend: transformer-based methods consistently outperform their MLP-based counterparts. In particular, DDPM-GPT achieves an average 21% improvement in success rate across Avoiding, Aligning, and Pushing tasks compared to DDPM-MLP. While DDPM-MLP exhibits higher entropy on the Aligning task, it lags behind on other tasks. When comparing VAE-MLP to VAE-ACT, VAE-ACT demonstrates a success rate improvement of over 30% on most tasks, indicating its effectiveness in capturing diverse behaviors. More significantly, transformers that incorporate historical inputs or prediction horizons demonstrate the ability to tackle challenging tasks like Sorting-4 and Stacking. However, the combination of historical information with extended prediction horizons (DDPM-ACT) does not appear to provide substantial benefits compared to DDPM-GPT, which does not predict future actions.

**Scaling to More Complex Tasks.** Current state-of-the-art methods excel in tasks such as Avoiding, Aligning, and Pushing; however, a notable performance gap emerges in Sorting and Stacking tasks. In the case of Sorting, scaling with an increasing number of objects proves challenging for all methods. Specifically, on the Sorting-6 task, none of the existing methods achieve satisfactory performance. The complexity of the observation space and task diversity significantly increases as we aggregate all box features into a single state vector. Consequently, there is a demand for models capable of learning compact and resilient state representations.

## 4.4 BENCHMARK USING IMAGE OBSERVATIONS

We present the results utilizing image observations in Table 3. Detailed information regarding camera setup can be found in Appendix A. Through an analysis of the performance and diversity across all methods, we outline our key findings below.

**Comparison between State-Based and Image-Based Policies.** State representations have proven highly effective for tasks demanding precise control (i.e. Aligning), but they do not scale with the number of objects (i.e. Sorting). In such case, state-based approaches may struggle to exploit invariances, which are essential for handling complex scenarios. Conversely, image representations exhibit a notable capacity to handle scenarios involving multiple objects. However, image-based policy performs much worse than state-based policy on the Aligning task, which require precise control. The inherent difficulty in extracting fine-grained details and spatial relationships from images makes it challenging in achieving precise manipulation.

**Trade-off in Sequential Information Across Visual Tasks.** From the results of state-based evaluations, it is evident that historical inputs and prediction horizons consistently enhance success rate and entropy on all tasks. Regarding the image-based results, DDPM-GPT shows less improvement than DDPM-MLP on Aligning and Sorting-4 and performs a little worse on Sorting-6. This phenomenon has also been observed in the comparison between BC-GPT and BC-MLP, BeT and BeT-MLP. However, in the most demanding task, Stacking, transformer-based models consistently outperform MLP

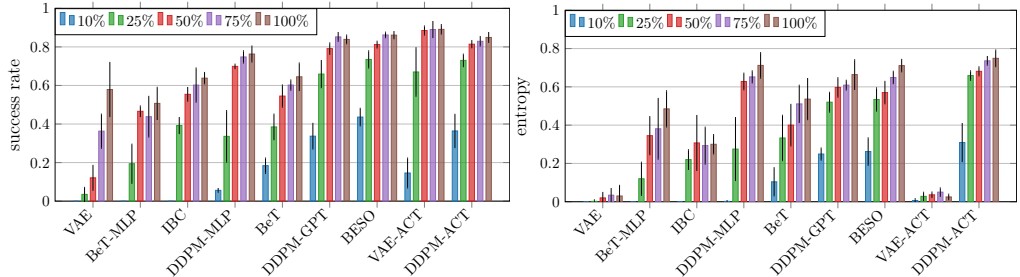

Figure 2: Ablation study for different percentages of the original dataset size of the Aligning (T2) task. The percentages are color-coded according to the legend in the top left corner.

models. Notably, DDPM-ACT displays improvements of 11%, 40%, 24% in stacking 1, 2, 3 boxes, respectively, compared to DDPM-MLP.

### 4.5    IMPACT OF HISTORY AND PREDICTION HORIZON

We conducted a comparison of various choices for history and prediction horizons, as illustrated in Figure 1. Notably, when both history and prediction horizons are greater than 1, DDPM-ACT exhibits improved performance. However, it's worth noting that, consistent with findings from prior work (Chi et al., 2023), extending the length of observation history and action sequence horizon leads to a decline in both success rate and entropy. This observation underscores the importance of careful design when using a transformer encoder-decoder structure for history and prediction horizons. Additionally, we find that historical inputs significantly enhance the performance of GPT-based policies, and this improvement is consistent with increasing the history length.

### 4.6    LEARNING WITH LESS DATA

The process of recording demonstrations, particularly when involving human demonstrators, is often tedious. To that end, we assess the models' ability to learn with less training data, we generate four subsets comprising 10%, 25%, 50% and 75% demonstrations for the Aligning task. The results are reported in Figure 2, with detailed results available in Table 6. MLP-based methods (e.g. VAE-State, BeT-MLP) experience a significant performance drop when trained with less data. They display up to 54.4% drop in success rate and 43.7% drop in entropy on the 25% dataset, and are nearly non-functional on the 10% dataset. In contrast, transformer-based methods (e.g. DDPM-GPT, BeT) exhibit a higher tolerance for small amounts of data, with up to 33.3% success rate drop and 20.3% entropy drop on 25% dataset, maintaining more than 15% success rate on 10% dataset. Additionally, we find that BESO exhibits a 43% success rate on the 10% dataset. From these findings, we conclude that transformer architecture generalizes well with less training data and diffusion-based methods seem to be able to regularize the transformer to make it less data-hungry.

## 5    CONCLUSION

Our introduction of *Datasets with Diverse human Demonstrations for Imitation Learning (D3IL)*, addresses the critical need to evaluate a model's capability to learn multi-modal behavior. These environments incorporate human data, involve intricate sub-tasks, necessitate the manipulation of multiple objects, and require policies based on closed-loop sensory feedback. Collectively, these characteristics significantly enhance the diversity of behavior in D3IL, setting it apart from existing benchmarks that often lack one or more of these crucial elements.

To measure this diversity, we introduce practical metrics that offer valuable insights into a model's ability to acquire and replicate diverse behaviors. Through a comprehensive evaluation of state-of-the-art methods on our proposed task suite, our research illuminates the effectiveness of these methods in learning diverse behavior. This contribution not only guides current efforts but also provides a valuable reference for the development of future imitation learning algorithms.

## 6 ACKNOWLEDGMENTS

This work was supported by funding from the pilot program Core Informatics of the Helmholtz Association (HGF). NS and GN were supported by the Carl Zeiss Foundation under the project JuBot (Jung Bleiben mit Robotern). Xiaogang Jia and Xinkai Jiang acknowledge the support from the China Scholarship Council (CSC). The authors acknowledge support by the state of Baden-Württemberg through bwHPC, as well as the HoreKa supercomputer funded by the Ministry of Science, Research and the Arts Baden-Württemberg and by the German Federal Ministry of Education and Research.

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

# A    ADDITIONAL TASK DETAILS

We provide additional visualizations and more detailed task descriptions.

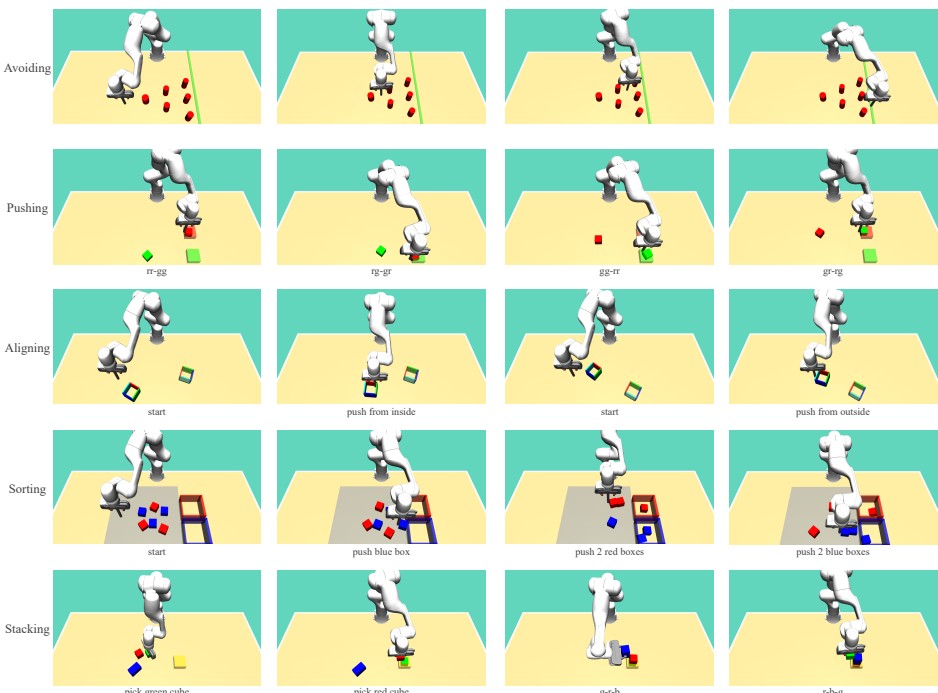

Figure 3: **D3IL Visualizations.** This figure provides an overview of various tasks and behaviors within our dataset. The top row demonstrates one of the 24 possible solutions for the "Avoiding" task. The second row displays snapshots from all four pushing sequences in the "Pushing" task, with sub-captions indicating block movements (e.g., 'rr-gg' signifies red block to red target and green block to green target). The third row showcases the two behaviors for the "Aligning" task, with the leftmost figures illustrating alignment from within the box and the rightmost from outside. The fourth row focuses on the "Sorting-3" task, with the initial configuration on the left and diverse pushing strategies, including simultaneous block manipulation, in subsequent figures. The bottom row depicts snapshots of the "Stacking" task, highlighting the intricate dexterity required, including complex pose estimation and orientation changes when picking and stacking the blue block.

## A.1    AVOIDING (T1)

**Success Criteria.** The *Avoiding* task is considered to be successful if the robot-end-effector reaches the green finish line without having a collision with one of the six obstacles.

**State and Action Representation.** The state representation encompasses the end-effector's desired position and actual position in Cartesian space, with the caveat that the robot's height remains fixed, resulting in $s \in \mathbb{R}^4$. The actions $a \in \mathbb{R}^2$ are represented by the desired velocity of the robot.

**Behavior Descriptors $\beta$ and Behavior Space $\mathcal{B}$.** The behavior descriptors $\beta$ defined by all 24 ways of avoiding obstacles and thus $|\mathcal{B}| = 24$. We collected data such that an equal amount of trajectories for each solution strategy exists.

## A.2 ALIGNING (T2)

**Success Criteria.** The *Aligning* task is considered to be successful if the box aligns with the target box within some tolerance and their colors match for each side.

**State and Action Representation.** The state representation includes the end-effector's desired position, actual position in Cartesian space, the pushing box position and quaternion, the target box position, and quaternion, and thus $\mathbf{s} \in \mathbb{R}^{20}$. The actions $\mathbf{a} \in \mathbb{R}^3$ are represented by the desired Cartesian velocity of the robot.

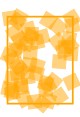 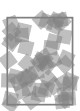

Figure 4: **Initial State Space.** The initial state $\mathbf{s}_0$ consists of the box and target position. The former is sampled uniformly from the yellow rectangle and the latter from the grey. The figure maintains the true-to-scale ratio of the samples and the bounding box.

**Behavior Descriptors $\beta$ and Behavior Space $\mathcal{B}$.** The behavior descriptors $\beta$ are conditioned on the initial state $\mathbf{s}_0$ and are defined by both ways of pushing a box, i.e., either from the inside or from the outside (refer to the Figure 3 Aligning). The data is recorded such that there are an approximately equal amount of trajectories for both behaviors and every initial state.

## A.3 PUSHING (T3)

**Success Criteria.** The *Pushing* task is deemed successful when both blocks are positioned inside a predefined target zone, with a certain level of tolerance allowed.

**State and Action Representation.** The state representation includes the end-effector's desired position, actual position in Cartesian space, the two boxes' position, and tangent of Euler angle along $z$ axis and thus $\mathbf{s} \in \mathbb{R}^{10}$. The actions $\mathbf{a} \in \mathbb{R}^2$ are represented by the desired Cartesian velocity of the robot with a fixed height.

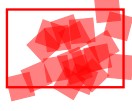
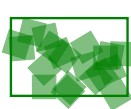

Figure 5: **Initial State Space.** The initial state $\mathbf{s}_0$ consists of the initial position of the red and green blocks. The red blocks are uniformly sampled from the red rectangle and the green ones from the green rectangle. The figure maintains the true-to-scale ratio of the samples and the bounding box.

**Behavior Descriptors $\beta$ and Behavior Space $\mathcal{B}$.** The behavior descriptors $\beta$ are conditioned on the initial state $\mathbf{s}_0$ and are defined by the four pushing sequences and thus $|\mathcal{B}| = 4$. The four pushing sequences are all combinations of pushing the two blocks to the target zones (refer to Figure 3 Pushing). The data is recorded such that there are an equal amount of trajectories for all pushing sequences and initial states.

## A.4 SORTING-X (T4)

**Success Criteria.** The *Sorting* task is considered successful when all blocks are positioned inside a target box with the same color.

**State and Action Representation.** The state representation includes the end-effector's desired position, actual position in Cartesian space, the boxes' position, and tangent of Euler angle along $z$ axis and thus $\mathbf{s} \in \mathbb{R}^{4+3X}$. The actions $\mathbf{a} \in \mathbb{R}^2$ are represented by the desired Cartesian position of the robot with a fixed height.

**Behavior Descriptors $\beta$ and Behavior Space $\mathcal{B}$.** The behavior descriptors $\beta$ are conditioned on the initial state $\mathbf{s}_0$ and are defined by all possible color combinations of sorting the blocks in the

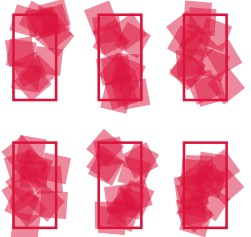

Figure 6: **Initial State Space.** The initial state $\mathbf{s}_0$ consists of the initial positions of all red and blue blocks. In case of collisions between blocks, we define 6 non-intersecting sampling spaces and uniformly sample 6 box's states which are randomly assigned to red and blue blocks. The figure maintains the true-to-scale ratio of the samples and the bounding box.

respective target zones. The data is recorded such that there are an approximately equal amount of trajectories for all sorting orders and every initial state.

## A.5 STACKING (T5)

**Success Criteria.** The *Stacking* task is deemed successful when all three blocks are in the yellow target zone (with some tolerance) and have the appropriate height, signifying that they are stacked on top of each other.

**State and Action Representation.** The state representation includes the robot's desired joint position, the width of the gripper, the boxes' position, and tangent of the Euler angle along $z$ axis and thus $\mathbf{s} \in \mathbb{R}^{20}$. The actions $\mathbf{a} \in \mathbb{R}^8$ are represented by the robot's desired velocity and width of the gripper.

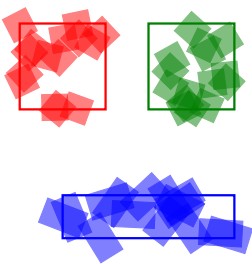

Figure 7: **Initial State Space.** The initial state $\mathbf{s}_0$ consists of the initial positions of red, green and blue blocks. The blocks are uniformly sampled from the corresponding rectangle box. The figure maintains the true-to-scale ratio of the samples and the bounding box.

**Behavior Descriptors $\beta$ and Behavior Space $\mathcal{B}$.** The behavior descriptors are conditioned on the initial state $\mathbf{s}_0$ and are defined as all possible color combinations of stacking the blocks in the respective target zones, resulting in $|\mathcal{B}| = 6$. The data is recorded such that there are an approximately equal amount of trajectories for all sorting orders and every initial state $\mathbf{s}_0$.

## A.6 ADDITIONAL DETAILS

**Datasets.** We present the statistics of our D3IL dataset. From T1 to T5, task complexity is increasing while the demonstrations require more time to record, with 61 steps minimum to 965 steps maximum. A large amount of dataset is provided for each task, with more than 1000 demonstrations for most tasks.

**Policy Simulation.** For each task, we uniformly sample $S_0$ initial states. Given an initial state, we simulate each policy in a closed-loop manner for multiple times based on task complexity. The policy is expected to learn diverse solutions for each context. We design the simulation time according to the maximum length of our demonstrations.

**State / Observation Representation.** For most experiments, we provide state representation and image observations. For the former, we use carefully handcrafted features that work well empirically. For the latter, we used two different camera views, in-hand and front view with a $96 \times 96$ image resolution as shown in Figure 8. We use 3 8-bit color channels normalized to $[0, 1]$ and a $96 \times 96$ image resolution, resulting in $\mathcal{S} = [0, 1]^{2 \times 3 \times 96 \times 96}$. We use a ResNet-18 architecture as an

| Tasks | $T_{\mu\pm\sigma}$ | $T_{\min}$ | $T_{\max}$ | $N_\tau$ | $N_{(\mathbf{s},\mathbf{a})}$ | $S_0$ | $N_{\text{sim}}$ |
|---|---|---|---|---|---|---|---|
| (T1) Avoiding | $77.09_{\pm 11.96}$ | 61 | 107 | 96 | 7.4k | – | 480 |
| (T2) Aligning | $194.88_{\pm 40.27}$ | 102 | 354 | 1000 | 194k | 60 | $8S_0$ |
| (T3) Pushing | $232.69_{\pm 31.19}$ | 152 | 356 | 2000 | 465k | 30 | $16S_0$ |
| (T4) Sorting-1 | $335.49_{\pm 32.62}$ | 112 | 297 | 600 | 112k | 60 | $8S_0$ |
| (T4) Sorting-2 | $335.49_{\pm 66.74}$ | 144 | 509 | 1054 | 353k | 60 | $18S_0$ |
| (T4) Sorting-3 | $510.57_{\pm 134.13}$ | 186 | 923 | 1560 | 848k | 60 | $20S_0$ |
| (T5) Stacking | $628.90_{\pm 73.33}$ | 458 | 965 | 1095 | 688k | 60 | $18S_0$ |

Table 4: Overview of Tasks. For each task, we report the demonstration steps $T$, number of demonstrations $N_\tau$, number of state-action pairs $N_{(\mathbf{s},\mathbf{a})}$, number of simulated initial states $S_0$ and number of simulations $N_{\text{sim}}$.

image encoder for all methods proposed in Diffusion Policy (Chi et al., 2023). For further details on the dimensionality of the state and action space for different tasks, see Table 5.

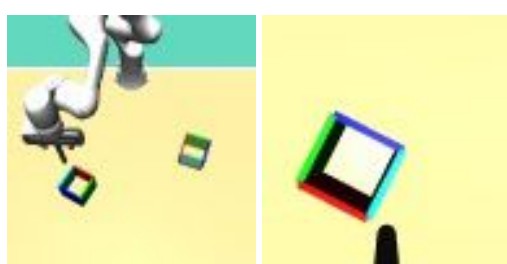

| Tasks | dim($\mathcal{S}$) | | dim($\mathcal{A}$) |
|---|---|---|---|
| | state rep. | image obs. | |
| Avoiding | 4 | - | 2 |
| Aligning | 20 | $(2, 3, 96, 96)$ | 3 |
| Pushing | 10 | - | 2 |
| Sorting-X | $4 + 3X$ | $(2, 3, 96, 96)$ | 2 |
| Stacking | 20 | $(2, 3, 96, 96)$ | 8 |

Figure 8: Camera views. Front view (right) and in-hand (left). The image resolution is $96 \times 96$.

Table 5: Summary of state-space dimensions dim($\mathcal{S}$) and action-space dimensions dim($\mathcal{A}$).

## B  FURTHER EXPERIMENTAL RESULTS

**Datasetsize Ablation Study.** We provide the exact numbers for the dataset size ablation study presented in Section 4.6, Figure 2. These numbers are presented in Table 6.

| | 10% dataset | | 25% dataset | | 50% dataset | | 75% dataset | | 100% dataset | |
|---|---|---|---|---|---|---|---|---|---|---|
| | Success Rate | entropy | Success Rate | entropy | Success Rate | entropy | Success Rate | entropy | Success Rate | entropy |
| BC-MLP | $0.0_{\pm 0.0}$ | $0.0_{\pm 0.0}$ | $0.208_{\pm 0.125}$ | $0.0_{\pm 0.0}$ | $0.586_{\pm 0.078}$ | $0.0_{\pm 0.0}$ | $0.691_{\pm 0.066}$ | $0.0_{\pm 0.0}$ | $0.708_{\pm 0.052}$ | $0.0_{\pm 0.0}$ |
| VAE-State | $0.0_{\pm 0.0}$ | $0.0_{\pm 0.0}$ | $0.035_{\pm 0.034}$ | $0.002_{\pm 0.005}$ | $0.121_{\pm 0.062}$ | $0.020_{\pm 0.026}$ | $0.363_{\pm 0.086}$ | $0.034_{\pm 0.032}$ | $0.579_{\pm 0.138}$ | $0.030_{\pm 0.053}$ |
| BeT-MLP | $0.0_{\pm 0.0}$ | $0.0_{\pm 0.0}$ | $0.194_{\pm 0.099}$ | $0.120_{\pm 0.084}$ | $0.466_{\pm 0.025}$ | $0.345_{\pm 0.097}$ | $0.438_{\pm 0.103}$ | $0.381_{\pm 0.156}$ | $0.507_{\pm 0.081}$ | $0.485_{\pm 0.092}$ |
| IBC | $0.0_{\pm 0.0}$ | $0.0_{\pm 0.0}$ | $0.392_{\pm 0.039}$ | $0.220_{\pm 0.049}$ | $0.554_{\pm 0.033}$ | $0.307_{\pm 0.141}$ | $0.602_{\pm 0.086}$ | $0.293_{\pm 0.094}$ | $0.638_{\pm 0.027}$ | $0.300_{\pm 0.048}$ |
| DDPM-MLP | $0.056_{\pm 0.007}$ | $0.002_{\pm 0.006}$ | $0.336_{\pm 0.131}$ | $0.275_{\pm 0.162}$ | $0.699_{\pm 0.009}$ | $0.628_{\pm 0.041}$ | $0.748_{\pm 0.029}$ | $0.652_{\pm 0.029}$ | $0.763_{\pm 0.039}$ | $0.712_{\pm 0.064}$ |
| BC-GPT | $0.163_{\pm 0.062}$ | $0.0_{\pm 0.0}$ | $0.500_{\pm 0.088}$ | $0.0_{\pm 0.0}$ | $0.722_{\pm 0.065}$ | $0.0_{\pm 0.0}$ | $0.811_{\pm 0.043}$ | $0.0_{\pm 0.0}$ | $0.833_{\pm 0.043}$ | $0.0_{\pm 0.0}$ |
| BeT | $0.184_{\pm 0.037}$ | $0.104_{\pm 0.071}$ | $0.385_{\pm 0.064}$ | $0.333_{\pm 0.116}$ | $0.545_{\pm 0.055}$ | $0.400_{\pm 0.106}$ | $0.602_{\pm 0.024}$ | $0.511_{\pm 0.095}$ | $0.645_{\pm 0.069}$ | $0.536_{\pm 0.105}$ |
| DDPM-GPT | $0.337_{\pm 0.064}$ | $0.249_{\pm 0.029}$ | $0.659_{\pm 0.068}$ | $0.520_{\pm 0.049}$ | $0.791_{\pm 0.027}$ | $0.597_{\pm 0.048}$ | $0.852_{\pm 0.020}$ | $0.609_{\pm 0.023}$ | $0.839_{\pm 0.020}$ | $0.664_{\pm 0.075}$ |
| BESO | $0.436_{\pm 0.043}$ | $0.262_{\pm 0.069}$ | $0.735_{\pm 0.042}$ | $0.534_{\pm 0.058}$ | $0.812_{\pm 0.015}$ | $0.570_{\pm 0.056}$ | $0.862_{\pm 0.012}$ | $0.649_{\pm 0.030}$ | $0.861_{\pm 0.016}$ | $0.711_{\pm 0.030}$ |
| VAE-ACT | $0.146_{\pm 0.075}$ | $0.006_{\pm 0.007}$ | $0.670_{\pm 0.123}$ | $0.028_{\pm 0.019}$ | $0.885_{\pm 0.021}$ | $0.036_{\pm 0.012}$ | $0.890_{\pm 0.039}$ | $0.050_{\pm 0.0194}$ | $0.891_{\pm 0.022}$ | $0.025_{\pm 0.012}$ |
| DDPM-ACT | $0.364_{\pm 0.084}$ | $0.309_{\pm 0.097}$ | $0.730_{\pm 0.030}$ | $0.659_{\pm 0.022}$ | $0.814_{\pm 0.016}$ | $0.681_{\pm 0.021}$ | $0.829_{\pm 0.022}$ | $0.736_{\pm 0.020}$ | $0.849_{\pm 0.023}$ | $0.749_{\pm 0.041}$ |

Table 6: Exact numbers for the dataset size ablation study presented in Section 4.6, Figure 2.

**Hyparameter Ablation Study.** To gain deeper insights into the hyperparameter sensitivity of our methods, we conducted an ablation study on the Aligning (T2) task, that highlights the influence of key hyperparameters on performance metrics. Specifically, for DDPM-GPT and BESO, we ablated the number of diffusion steps; for BeT, we focused on the number of action clusters, and for VAE-ACT, we ablated the scaling factor of the Kullback-Leibler divergence in the loss term. The detailed results are presented in Figure 9.

Our findings reveal that increasing the number of diffusion steps primarily enhances behavior entropy, signifying an improvement in the model's capacity to capture multi-modalities. However, for BeT and VAE-ACT, we did not observe a consistent trend indicating a correlation between the parameters and success rate or entropy.

**Inference Time Comparison.** We conducted a comprehensive comparison of inference times for all baseline models using both state- and image-based inputs. For state-based inputs, we utilized a

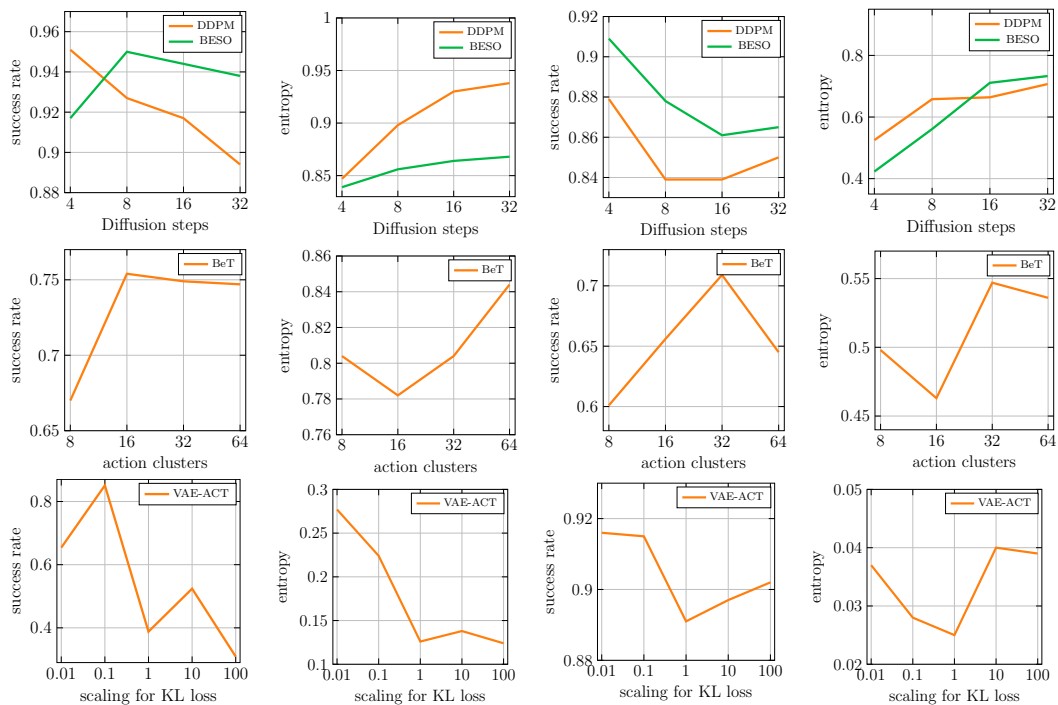

Figure 9: Ablation Study for the most important hyperparameters of DDPM/BESO (diffusion steps), BeT (action clusters), and VaE (scaling factor for Kullback-Leibler loss term) on the Avoiding (T1) and Aligning (T2) task.

ten-dimensional input, while for images, a $2 \times 3 \times 96 \times 96$ dimensional input was employed. To ensure robustness, we repeated the inference process 500 times and calculated the average time. The detailed results are presented in Table 7.

It is evident that methods employing (discretized) Langevin dynamics, such as IBC and diffusion-based models, exhibit longer inference times. It's noteworthy that methods labeled with 'ACT' predict three steps into the future, resulting in lower per-step inference times.

|  | Inference Time | |
|---|---|---|
|  | state-based | image-based |
| BC-MLP | 0.46 ms | 5.77 ms |
| VAE-State | 0.52 ms | 6.41 ms |
| BeT-MLP | 1.00 ms | 6.56 ms |
| IBC | 21.94 ms | 26.53 ms |
| DDPM-MLP | 5.29 ms | 11.53 ms |
| VAE-ACT | 0.64 ms | 5.68 ms |
| BC-GPT | 1.84 ms | 7.82 ms |
| BeT | 1.97 ms | 7.89 ms |
| DDPM-GPT | 30.52 ms | 43.56 ms |
| BESO | 31.14 ms | 40.30 ms |
| DDPM-ACT | 12.75 ms | 14.81 ms |

Table 7: Inference time comparison averaged over 500 runs. Methods labeled with 'ACT' predict three steps into the future, resulting in lower per-step inference times.

**Position vs. Velocity Control.** Similar to Chi et al. (2023), we conducted a comparison between position control (absolute actions) and velocity control (relative actions) for DDPM-ACT and BESO across three distinct tasks. The results, obtained from four different seeds, are presented in Table 8.

Our findings indicate that employing velocity control leads to a significant improvement in both success rate and entropy for the tasks.

| | Avoiding (T1) | | Aligning (T2) | | Pushing (T3) | |
|---|---|---|---|---|---|---|
| Velocity Control | Success Rate | Entropy | Success Rate | Entropy | Success Rate | Entropy |
| BESO | $0.950_{\pm 0.015}$ | $0.856_{\pm 0.018}$ | $0.861_{\pm 0.016}$ | $0.711_{\pm 0.030}$ | $0.794_{\pm 0.095}$ | $0.871_{\pm 0.037}$ |
| DDPM-ACT | $0.899_{\pm 0.006}$ | $0.863_{\pm 0.068}$ | $0.849_{\pm 0.023}$ | $0.749_{\pm 0.041}$ | $0.920_{\pm 0.027}$ | $0.859_{\pm 0.012}$ |
| Position Control | Success Rate | Entropy | Success Rate | Entropy | Success Rate | Entropy |
| BESO | $0.459_{\pm 0.333}$ | $0.327_{\pm 0.226}$ | $0.133_{\pm 0.024}$ | $0.071_{\pm 0.066}$ | $0.013_{\pm 0.010}$ | $0.000_{\pm 0.000}$ |
| DDPM-ACT | $0.153_{\pm 0.040}$ | $0.836_{\pm 0.054}$ | $0.413_{\pm 0.163}$ | $0.385_{\pm 0.347}$ | $0.404_{\pm 0.141}$ | $0.478_{\pm 0.077}$ |

Table 8: Comparison between position control (absolute actions) and velocity control (relative actions). The results are averaged over four seeds.

**Result Visualization for Avoiding Task (T1).** To offer deeper insights into the diverse behaviors of the policies, we present visualizations of the end-effector trajectories for various methods in Figure 10.

IBC, BeT, BESO, and DDPM-ACT manage to acquire a diverse set of skills, while the other methods struggle to represent the different modes present in the data distribution. This is consistent with the results presented in Table 3.

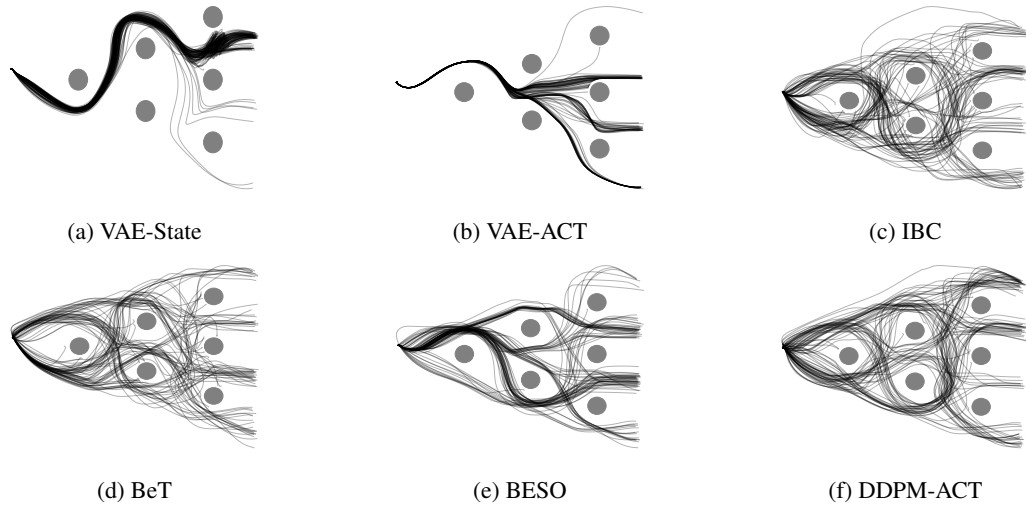

(a) VAE-State     (b) VAE-ACT     (c) IBC

(d) BeT     (e) BESO     (f) DDPM-ACT

Figure 10: Visualization of 100 end-effector trajectories for different methods, as indicated by the sub-captions.

## C    ADDITIONAL BASELINE DETAILS

We compare the following state-of-the-art algorithms in learning multi-modal behavior from human demonstrations:

**Behavior Cloning (BC)**. Following the traditional behavior cloning, MLP is optimized by Mean Squared Error (MSE) which assumes the underlying data distribution is unimodal Gaussian. As a result, MLP is not able to capture diverse behaviors. BC is used as a default baseline for the success rate.

**State Variational Autoencoder-Policies (VAE-State)**. We evaluate a State-VAE (Sohn et al., 2015; Kingma & Welling, 2013), that encodes current observation and action into a shared latent space and uses a state-conditioned decoder to reconstruct the actions.

**Action Chunking Variational Autoencoder-Policies (VAE-ACT)**. An action sequence VAE model (Zhao et al., 2023; Bharadhwaj et al., 2023), that encodes a sequence of actions into a small latent space $z = f(a_{t:t+k})$ with an VAE and conditions an transformer policy to reconstruct the action sequence.

**Behavior Transformers (BeT)**. (Shafiullah et al., 2022) BeT uses K-mean discretization to learn K-discrete action centers from the dataset together with an offset vector. We further evaluate an BeT variant using an MLP ablation to disentangle the contributions of the transformer architecture and the discrete action representation.

**Implicit Behavioral Cloning (IBC)**. (Florence et al., 2022) The policy is represented with an implicit energy-based model (EBM). Actions are generated using gradient-based Langevin sampling during inference. EBMs have shown to effectively represent multi-modal behavior using their implicit representation.

**Discrete Diffusion Policies (DDPM)**. (Pearce et al., 2023; Chi et al., 2023) We further evaluate several diffusion policies We represent the policy using a conditional discrete Diffusion model (Ho et al., 2020), that learns to reverse a diffusion process. New action samples are generated in an iterative denoising method starting from pure random Gaussian noise. We evaluate diffusion policies with every architecture.

**Continuous Time Diffusion Policies (BESO)**. (Reuss et al., 2023) Another diffusion-based policy, that represents the denoising process using a continuous stochastic-differential equation (SDE) (Song et al., 2020) instead of using a discrete time markov process.

We tuned the most important hyperparameters of these methods using Bayesian optimization (Snoek et al., 2012). A detailed summary of our hyperparameters can be found in Table 9.

| Methods / Parameters | Grid Search | Avoiding | Aligning | Pushing | Sorting-2 | Sorting-4 | Sorting-6 | Stacking | Inserting | Arranging |
|---|---|---|---|---|---|---|---|---|---|---|
| **BC-MLP** | | | | | | | | | | |
| Learning Rate $\times 10^{-4}$ | $\{1, 5, 10, 50\}$ | 10 | 10 | 10 | 10 | 10 | 10 | 10 | 10 | 10 |
| Hidden Layer Dimension | $\{64, 128, 256\}$ | 128 | 128 | 128 | 128 | 256 | 256 | 256 | 128 | 128 |
| Number of Hidden Layers | $\{4, 6, 8, 10\}$ | 6 | 6 | 6 | 6 | 8 | 8 | 8 | 6 | 6 |
| **VAE** | | | | | | | | | | |
| Learning Rate $\times 10^{-4}$ | $\{1, 5, 10, 50\}$ | 1 | 1 | 1 | 1 | 1 | 1 | 1 | 1 | 1 |
| Latent Dimension | $\{8, 16, 32, 64\}$ | 32 | 16 | 24 | 24 | 16 | 16 | 16 | 16 | 16 |
| KL Loss Scaling | $[0.001, 100.0]$ | 59.87 | 86.03 | 62.86 | 38.87 | 67.46 | 67.46 | 67.46 | 67.46 | 67.46 |
| **BeT-MLP** | | | | | | | | | | |
| Learning Rate $\times 10^{-4}$ | $\{1, 5, 10, 50\}$ | 1 | 1 | 1 | 1 | 1 | 1 | 1 | 1 | 1 |
| Number of Bins | $\{16, 24, 32, 64\}$ | 64 | 64 | 16 | 24 | 64 | 64 | 64 | 64 | 64 |
| Offset Loss Scale | $\{1.0, 10.0, 100.0, 1000.0\}$ | 1.0 | 1.0 | 1.0 | 1.0 | 1.0 | 1.0 | 1.0 | 1.0 | 1.0 |
| **IBC** | | | | | | | | | | |
| Learning Rate $\times 10^{-4}$ | $\{1, 5, 10, 50\}$ | 10 | 10 | 10 | 10 | 10 | 10 | 10 | 10 | 10 |
| Train Iterations | $\{10, 20, 40, 50\}$ | 40 | 40 | 40 | 40 | 40 | 40 | 40 | 40 | 40 |
| Inference Iterations | $\{10, 20, 30\}$ | 10 | 10 | 10 | 10 | 10 | 10 | 10 | 10 | 10 |
| Sampler Step Size initialization | $[0.01, 0.8]$ | 0.0493 | 0.0493 | 0.0493 | 0.0493 | 0.0493 | 0.0493 | 0.0493 | 0.0493 | 0.0493 |
| **DDPM-MLP** | | | | | | | | | | |
| Learning Rate $\times 10^{-4}$ | $\{1, 5, 10, 50\}$ | 10 | 10 | 10 | 10 | 10 | 10 | 10 | 10 | 10 |
| Number of Time Steps | $\{4, 8, 16, 24, 32, 64\}$ | 4 | 24 | 8 | 4 | 4 | 4 | 4 | 4 | 4 |
| Time Step Embedding | $\{4, 8, 16, 24\}$ | 24 | 8 | 16 | 4 | 8 | 4 | 4 | 4 | 4 |
| **VAE-ACT** | | | | | | | | | | |
| Learning Rate $\times 10^{-4}$ | $\{1, 5, 10, 50\}$ | 1 | 1 | 1 | 1 | 1 | 1 | 1 | 1 | 1 |
| Encoder Layers | − | 2 | 2 | 2 | 2 | 2 | 2 | 2 | 2 | 2 |
| Encoder Heads | − | 4 | 4 | 4 | 4 | 4 | 4 | 4 | 4 | 4 |
| Encoder Embedding | $\{64, 72, 120\}$ | 64 | 64 | 64 | 64 | 64 | 64 | 64 | 64 | 64 |
| Decoder Layers | − | 4 | 4 | 4 | 4 | 4 | 4 | 4 | 4 | 4 |
| Decoder Heads | − | 4 | 4 | 4 | 4 | 4 | 4 | 4 | 4 | 4 |
| Decoder Embedding | $\{64, 72, 120\}$ | 64 | 64 | 64 | 64 | 64 | 64 | 64 | 64 | 64 |
| Prediction Horizon | $\{3, 5, 8, 10\}$ | 3 | 3 | 3 | 3 | 3 | 3 | 3 | 3 | 3 |
| Latent Dimension | $\{8, 16, 32, 64\}$ | 32 | 32 | 32 | 32 | 32 | 32 | 32 | 32 | 32 |
| KL Loss Scaling | $[0.001, 10.0]$ | 0.1 | 1.0 | 1.0 | 1.0 | 1.0 | 1.0 | 1.0 | 1.0 | 1.0 |
| **BC-GPT** | | | | | | | | | | |
| Learning Rate $\times 10^{-4}$ | $\{1, 5, 10, 50\}$ | 1 | 1 | 1 | 1 | 1 | 1 | 1 | 1 | 1 |
| Number of Layers | − | 4 | 4 | 4 | 4 | 6 | 6 | 6 | 4 | 4 |
| Number of Attention Heads | − | 4 | 4 | 4 | 4 | 6 | 120 | 120 | 4 | 4 |
| Embedding Dimension | − | 72 | 72 | 72 | 72 | 120 | 120 | 120 | 72 | 72 |
| **BeT** | | | | | | | | | | |
| Learning Rate $\times 10^{-4}$ | $\{1, 5, 10, 50\}$ | 1 | 1 | 1 | 1 | 1 | 1 | 1 | 1 | 1 |
| Number of Bins | $\{8, 16, 32, 64\}$ | 64 | 64 | 16 | 24 | 64 | 64 | 64 | 64 | 64 |
| Offset Loss Scale | $\{1.0, 10.0, 100.0, 1000.0\}$ | 1.0 | 1.0 | 1.0 | 1.0 | 1.0 | 1.0 | 1.0 | 1.0 | 1.0 |
| **DDPM-GPT** | | | | | | | | | | |
| Learning Rate $\times 10^{-4}$ | $\{1, 5, 10, 50\}$ | 5 | 5 | 5 | 5 | 5 | 5 | 5 | 5 | 5 |
| Number of Time Steps | $\{4, 8, 16, 32, 64\}$ | 8 | 16 | 64 | 16 | 16 | 16 | 16 | 16 | 16 |
| **BESO** | | | | | | | | | | |
| Learning Rate $\times 10^{-4}$ | $\{1, 5, 10, 50\}$ | 5 | 5 | 5 | 5 | 5 | 5 | 5 | 5 | 5 |
| Number of Sampling Steps | $\{4, 8, 16, 32, 64\}$ | 8 | 16 | 64 | 16 | 16 | 16 | 16 | 16 | 16 |
| $\sigma_{min}$ | $\{0.001, 0.005, 0.01, 0.05, 0.1\}$ | 0.1 | 0.01 | 0.1 | 0.1 | 0.1 | 0.1 | 0.1 | 0.1 | 0.1 |
| $\sigma_{max}$ | $\{1, 3, 5\}$ | 1 | 3 | 1 | 1 | 1 | 1 | 1 | 1 | 1 |
| **DDPM-ACT** | | | | | | | | | | |
| Learning Rate $\times 10^{-4}$ | $\{1, 5, 10, 50\}$ | 5 | 5 | 5 | 5 | 5 | 5 | 5 | 5 | 5 |
| Encoder Layers | − | 2 | 2 | 2 | 2 | 2 | 2 | 2 | 2 | 2 |
| Encoder Heads | − | 4 | 4 | 4 | 4 | 4 | 4 | 4 | 4 | 4 |
| Encoder Embedding | $\{64, 72, 120\}$ | 64 | 64 | 64 | 64 | 120 | 120 | 120 | 64 | 64 |
| Decoder Layers | − | 4 | 4 | 4 | 4 | 4 | 4 | 4 | 4 | 4 |
| Decoder Heads | − | 4 | 4 | 4 | 4 | 4 | 4 | 4 | 4 | 4 |
| Decoder Embedding | $\{64, 72, 120\}$ | 64 | 64 | 64 | 64 | 120 | 120 | 120 | 64 | 64 |
| Number of Sampling Steps | $\{4, 8, 16, 32, 64\}$ | 16 | 16 | 16 | 16 | 16 | 16 | 16 | 16 | 16 |
| History | $\{3, 5, 10\}$ | 5 | 5 | 5 | 5 | 5 | 5 | 5 | 5 | 5 |
| Prediction Horizon | $\{3, 5, 8, 10\}$ | 3 | 3 | 3 | 3 | 3 | 3 | 3 | 3 | 3 |

Table 9: Hyperparameter Selection for all Imitation Learning algorithms for D3IL experiments. The 'Grid Serach' column indicates the values over which we performed a grid search. The values in the column which are marked with task names indicate which values were chosen for the reported results.

# D   ADDITIONAL TASKS

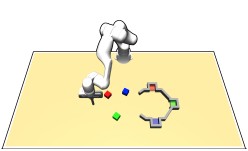

Figure 11: **Inserting (T6).** Similar to T3, the Inserting task involves the robot pushing blocks to designated target zones. However, this task is more complex due to: i) the presence of three blocks and corresponding target zones, resulting in $|\mathcal{B}| = 6$, ii) longer time horizons, and iii) the need for dexterous manipulations arising from additional constraints imposed by the gray barrier. The dataset comprises a total of 800 demonstrations, evenly distributed among the six behavior descriptors $\beta$.

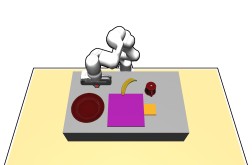

Figure 12: **Arranging (T7).** This task involves the robot arranging various objects on the table. Specifically, the robot is required to: i) flip the cup and place it in the yellow target zone, ii) position the banana on the plate, and iii) position the plate in the purple target zones. The diversity in this task is introduced by the order of these subtasks. The dataset consists of 585 demonstrations and is recorded using augmented reality (AR) such that each arranging order is equally often present.

| | Inserting (T6) | | Arranging (T7) | | | | |
| | Success Rate | Entropy | Banana | Plate | Cup | Success Rate | Entropy |
|---|---|---|---|---|---|---|---|
| BC-MLP | $0.025_{\pm 0.016}$ | $0.0_{\pm 0.0}$ | $0.0_{\pm 0.0}$ | $0.091_{\pm 0.142}$ | $0.0_{\pm 0.0}$ | $0.0_{\pm 0.0}$ | $0.0_{\pm 0.0}$ |
| VAE-State | $0.014_{\pm 0.008}$ | $0.006_{\pm 0.012}$ | $0.003_{\pm 0.004}$ | $0.025_{\pm 0.033}$ | $0.0_{\pm 0.0}$ | $0.0_{\pm 0.0}$ | $0.0_{\pm 0.0}$ |
| BeT-MLP | $0.021_{\pm 0.009}$ | $0.015_{\pm 0.012}$ | $0.011_{\pm 0.013}$ | $0.129_{\pm 0.066}$ | $0.009_{\pm 0.007}$ | $0.0_{\pm 0.0}$ | $0.0_{\pm 0.0}$ |
| IBC | $0.066_{\pm 0.018}$ | $0.125_{\pm 0.046}$ | $0.0_{\pm 0.0}$ | $0.0_{\pm 0.0}$ | $0.0_{\pm 0.0}$ | $0.0_{\pm 0.0}$ | $0.0_{\pm 0.0}$ |
| DDPM-MLP | $0.060_{\pm 0.018}$ | $0.108_{\pm 0.061}$ | $0.004_{\pm 0.004}$ | $0.092_{\pm 0.081}$ | $0.033_{\pm 0.047}$ | $0.0_{\pm 0.0}$ | $0.0_{\pm 0.0}$ |
| VAE-ACT | $0.300_{\pm 0.086}$ | $0.181_{\pm 0.010}$ | $0.108_{\pm 0.099}$ | $\mathbf{0.517_{\pm 0.397}}$ | $0.083_{\pm 0.137}$ | $0.020_{\pm 0.029}$ | $0.006_{\pm 0.012}$ |
| BC-GPT | $0.0_{\pm 0.0}$ | $0.0_{\pm 0.0}$ | $0.058_{\pm 0.041}$ | $0.441_{\pm 0.358}$ | $0.0_{\pm 0.0}$ | $0.0_{\pm 0.0}$ | $0.0_{\pm 0.0}$ |
| BeT | $0.466_{\pm 0.046}$ | $0.683_{\pm 0.018}$ | $0.036_{\pm 0.022}$ | $0.371_{\pm 0.291}$ | $0.046_{\pm 0.051}$ | $0.001_{\pm 0.001}$ | $0.0_{\pm 0.0}$ |
| DDPM-GPT | $0.112_{\pm 0.105}$ | $0.231_{\pm 0.241}$ | $0.180_{\pm 0.111}$ | $0.337_{\pm 0.089}$ | $0.418_{\pm 0.259}$ | $0.138_{\pm 0.096}$ | $\mathbf{0.101_{\pm 0.094}}$ |
| BESO | $0.058_{\pm 0.020}$ | $0.091_{\pm 0.045}$ | $\mathbf{0.253_{\pm 0.092}}$ | $0.271_{\pm 0.086}$ | $\mathbf{0.687_{\pm 0.110}}$ | $\mathbf{0.156_{\pm 0.057}}$ | $0.059_{\pm 0.027}$ |
| DDPM-ACT | $\mathbf{0.654_{\pm 0.088}}$ | $\mathbf{0.744_{\pm 0.021}}$ | $0.107_{\pm 0.054}$ | $0.306_{\pm 0.123}$ | $0.210_{\pm 0.056}$ | $0.054_{\pm 0.037}$ | $0.055_{\pm 0.045}$ |

Table 10: Results for the Inserting (T6) and Arranging (T7) task for state-based data. For T7, we additionally reported the success rate for sub-tasks, including i) placing the banana on the plate (Banana) ii) pushing the plate to the target area (Plate) and iii) flipping the cup and putting it in the target area (Cup).

