# OpenReview forum: "Towards Diverse Behaviors: A Benchmark for Imitation Learning with Human Demonstrations"
_ICLR.cc/2024/Conference — ICLR 2024 poster_

### Official Review · Reviewer_DMVD · 2023-10-30

**Soundness:** 3 good
**Presentation:** 3 good
**Contribution:** 3 good
**Rating:** 8
**Confidence:** 4

**Summary:**

The paper proposes a new benchmark for imitation learning with a focus on evaluating diverse behaviors. The authors perform an extensive comparison of different imitation learning algorithms, ranging from deterministic to stochastic algorithms using MLPs and transformers along with interesting insights about state-based and image-based policies and other algorithmic aspects.

**Strengths:**

- The paper introduces a new benchmark for imitation learning along with appropriate metrics to quantitatively evaluate diverse behaviors.
- The paper performs ablation studies and provides good insights about state-based vs image-based policies, the impact of history and action prediction horizon, and learning with less data.
- The paper provides some results that are consistent across methods - (1) transformers improve performance over using MLPs, (2) historical inputs enhance the performance of transformer-based policies, and (3) transformers exhibit superior performance in the low data regime.
- The paper has tasks of varying difficulty with a task like stacking-3 which is not satisfactorily solved by any of the existing algorithms. This provides a scope for improvement.

**Weaknesses:**

- Based on the results, it seems like all tasks can be solved by existing methods except one. Though this gives some scope for improvement on the algorithmic side, I believe just a single variant of a task remaining unsolved might not be a very useful for future works considering this benchmark for evaluations. It would be great if the authors could include other tasks or provide functionalities for adding new tasks.
- It would be great if the authors could provide code since a benchmark is only useful if the code is available.

**Questions:**

It would be great if the author’s could address the points mentioned in “Weaknesses”.

---

> ### Author Response · Authors · 2023-11-16
> **Rebuttal by Authors**
>
> We thank the reviewers for their feedback on our paper. We would like to address the mentioned weaknesses:
>
> >Based on the results, it seems like all tasks can be solved by existing methods except one. Though this gives some scope for improvement on the algorithmic side, I believe just a single variant of a task remaining unsolved might not be a very useful for future works considering this benchmark for evaluations.
>
> We kindly express some uncertainty regarding the specific definition the reviewer uses to characterize a task as solved. Assuming the criteria for considering a task as solved is a success rate exceeding 0.8, we acknowledge that two tasks, Sorting (T4) and Stacking (T5), remain unsolved for both state- and image-based policies in our study.
>
> It's important to highlight that one of the strengths of our work lies in introducing an additional performance metric, namely *behavior entropy*. This metric provides additional dimensions for improvement across tasks.
>
> ---
>
> >It would be great if the authors could include other tasks […]
>
> We appreciate the reviewer's suggestion and concur with the idea that the inclusion of more tasks enhances the quality of our work. Accordingly, we have introduced two additional tasks, *Inserting (T6) and Arranging (T7).*
>
> *Inserting (T6)* is conceived as a more challenging variant of the Pushing (T3) task, encompassing more blocks, increased behavior-level multi-modality, and demanding more dexterity. We have collected over 800 demonstrations using a gamepad controller. Preliminary results indicate that this task presents room for improvement across current methods.
>
> *Arranging (T7)* involves organizing objects on a table, such as flipping a cup. We are actively engaged in data collection utilizing our augmented reality system, and while preliminary results might not be available immediately, we aim to provide them during the rebuttal phase.
>
> We included task descriptions and the preliminary results in a revised version of our paper which are highlighted with blue color and can be found in Appendix D. We are committed to incorporating these tasks, along with a comprehensive evaluation of all methods, in the final camera-ready version of our work.
>
> ---
>
>
> >It would be great if the authors could provide code […]
>
> We have included the code base as a .zip file in the supplementary material. Additionally, we have provided a link to an anonymous GitHub repository, accessible here: https://github.com/d3iltest/Temporary_D3IL.
>
> ---
>
>
> >[…] or provide functionalities for adding new tasks.
>
> The code base is accompanied by a comprehensive readme. This readme provides detailed instructions on how to add new tasks to the framework. Specifically, it guides users through the process of setting up a new environment and recording data using a gamepad.
>
>
> We express our gratitude to the reviewer for their valuable comments and suggestions. We are pleased to address any additional questions or concerns that may arise.
>
> ---
>
> [1] Diffusion Policy: Visuomotor Policy Learning via Action Diffusion, RSS ‘23
>
> [2] Goal-Conditioned Imitation Learning using Score-based Diffusion Policies, RSS ‘23

---

> > ### Author Response · Authors · 2023-11-21
> > **Rebuttal by Authors (cont.)**
> >
> > Dear Reviewer DMVD,
> >
> > I'm pleased to share with you some preliminary results regarding the recently introduced tasks, Inserting (T6) and Arranging (T7). We collected over 800 demonstrations for T6 using a gamepad controller and nearly 600 demonstrations for T7 using our AR system. We employed three methods with proven success in other tasks to assess our new tasks, utilizing a state-based input representation. The current results indicate that none of the models achieved satisfactory performance, underscoring the potential for improvement in future research endeavors.
> >
> >
> > We have uploaded a revised version of the paper, incorporating the task descriptions along with the new results in Appendix D, marked with blue color for easy identification. Additionally, we have included the results in the tables below for your immediate reference.
> >
> > |      T6        | Success Rate                    | Entropy                         |
> > |--------------|---------------------------------|---------------------------------|
> > | VAE-ACT      | $0.251 \scriptstyle{\pm 0.039}$ | $0.301 \scriptstyle{\pm 0.058}$ |
> > | BeT          | $0.461 \scriptstyle{\pm 0.065}$ | $0.675 \scriptstyle{\pm 0.041}$ |
> > | DDPM-ACT     | $0.659 \scriptstyle{\pm 0.043}$ | $0.722 \scriptstyle{\pm 0.038}$ |
> >
> > |       T7       | Banana                          | Plate                           | Cup                             | Success Rate                    | Entropy                         |
> > |--------------|---------------------------------|---------------------------------|---------------------------------|---------------------------------|---------------------------------|
> > | VAE-ACT      | $0.154 \scriptstyle{\pm 0.023}$ | $0.025 \scriptstyle{\pm 0.011}$ | $0.004 \scriptstyle{\pm 0.001}$ | $0.0 \scriptstyle{\pm 0.0}$                                    | $0.0 \scriptstyle{\pm 0.0}$                             |
> > | BeT          | $0.064 \scriptstyle{\pm 0.056}$ | $0.297 \scriptstyle{\pm 0.054}$ | $0.050 \scriptstyle{\pm 0.034}$ | $0.0 \scriptstyle{\pm 0.0}$                                 | $0.0 \scriptstyle{\pm 0.0}$                           |
> > | DDPM-ACT     | $0.172 \scriptstyle{\pm 0.056}$ | $0.400 \scriptstyle{\pm 0.027}$ | $0.220 \scriptstyle{\pm 0.019}$ | $0.075 \scriptstyle{\pm 0.010}$ | $0.098 \scriptstyle{\pm 0.008}$ |
> >
> > For T7, we additionally reported the success rate for sub-tasks, including i) placing the banana on the plate (Banana) ii) pushing the plate to the target area (Plate) and iii) flipping the cup and putting it in the target area (Cup).
> >
> > ---
> >
> > Thank you for your suggestions for improving our paper. Here, we humbly ask for your feedback at your convenience such that we may still have the chance to discuss with you and incorporate improvements into our final version before the conclusion of the rebuttal phase.

---

> ### Comment · Reviewer_DMVD · 2023-11-22
> **Thank you for the rebuttal**
>
> I thank the authors for the rebuttal. My concerns have been sufficiently addressed and I am raising my score from 5 -> 8.

---

### Official Review · Reviewer_S4Qj · 2023-10-30

**Soundness:** 3 good
**Presentation:** 3 good
**Contribution:** 3 good
**Rating:** 6
**Confidence:** 4

**Summary:**

The authors propose a benchmarking for imitation learning from human demonstrations. Compared to other released benchmark datasets, the authors place emphasis on using human demonstrations and those demonstrations covering diverse behaviors.  The argument for doing so is because human demonstratinos inherently have some noise if the teleoperator differs, if people have different levels of expertise, etc. The goal is also to propose a quantitative measure of diverse behavior.

The proposal for this is to assume there exists a space $\mathcal{B}$ of discrete behavior descriptions $\beta \in \mathcal{B}$ (i.e. for pushing, whether we are pushing red to red or red to green and in what order). Our demonstrations define a $p(\beta)$ distribution of how often different behaviors appear. A learned policy $\pi$ will induces its own $\pi(\beta)$ distribution of behaviors, and similarity is measured by $KL(\pi(\beta) || p(\beta))$. This is simplified to a uniform distribution for $p(\beta)$ in all experiments, with entropy scaled to lie in range $[0,1]$. This reduces to $H(\pi) = - \sum \pi(\beta) \log_{|B|}\pi(\beta)$

Diversity is further defined as achieving many behaviors from the same initial state $s_0$, giving the conditional behavior entropy of

$$
H(\pi(\beta|s_0)) \approx -\frac{1}{S_0} \sum_{s_0} \sum_{\beta} \pi(\beta|s_0) \log_{|B|} \pi(\beta|s_0)
$$

Since the $p(s_0)$ distribution is unknown, this is approximated by Monte Carlo estimates using $S_0$ samples of the initial state.

The proposed benchmark is implemented in MuJoCo using a Panda robot and mostly consists of block moving tasks. A variety of imitation learning methods are tried, varying from pure MLPs to history-aware methods and diffusion policies. Experiments are also conduted on history length and on size of dataset.

**Strengths:**

The authors run an extensive series of benchmark methods over their proposed environments. The environments are described in sufficient detail to reproduce them. The evaluation protocol and model selection criteria is documented well, a rarity in robot learning papers. The paper is also written quite clearly.

**Weaknesses:**

The fact that different behaviors must be enumerated ahead of time is a large limitation of the proposed behavior-level entropy measure. I also found the KL-divergence to be a bit unmotivated. This paper assumes the demonstration dataset is always uniformly distributed among all behaviors, but in cases where the demonstration dataset is not uniformly distributed, it's not clear to me if KL divergence is the right measure to use. (We would expect a good learning method to have low KL, but if the demo dataset is skewed, we may still prefer a policy that is uniform across behaviors, even if this has higher KL than identicall matching the skewed distribution.) It seems like the entire discussion about KL is pointless and it would be more straightforward to just use the behavior entropy definition.

Setting aside this for the moment, the paper also does not ever make a claim that multimodal policies would be good. Success rate need not be correlated with high behavior entropy - as argued by the experiment results, deterministic policies can still achieve okay success rate without diverse behavior. And hypothetically, you could have a 100% success rate policy that only follows a singular behavior $\beta$. Such a policy may even be preferred (i.e. in factory automation, repeatable behavior given initial conditions is desired.)

Arguably, the paper is just about measuring this quantity, rather than arguing why it matters, but I would have appreciated some argument on this front.

**Questions:**

Overall I feel the paper is okay, despite the flaws it does make some strides towards focusing on diversity of behavior. But could the authors comment on where conditional imitation learning falls into the picture. In the pushing task for example, if the 4 behaviors are known ahead of time, you could imagine conditioning the policy on a 1-hot with 4 values for "push X1 to Y1, push X2 to Y2", and that would allow a deterministic policy to achieve any of the behaviors assuming a perfect learning method. What is the argument for why we cannot or should not do something where we provide additional context to the policy on how we want the task to be performed?

---

> ### Author Response · Authors · 2023-11-16
> **Rebuttal by Authors**
>
> We would like to thank the reviewer for taking the time to review our work and the many helpful comments and suggestions. We are committed to addressing your questions and concerns.
>
> ---
>
> > The fact that different behaviors must be enumerated ahead of time is a large limitation of the proposed behavior-level entropy measure.
>
> We acknowledge the reviewer's concern about the limitation imposed by the prerequisite to enumerate different behaviors in advance for the proposed behavior-level entropy measure. However, quantifying diverse behavior is a challenging problem, for which, to the best of our knowledge, exists no approach that circumvents this requirement. Moreover, there exists no task suite that provides any metric for quantifying diverse behaviors as highlighted in Table 1 of the paper. In light of this, we believe that our work makes a valuable contribution to the field.
>
> ---
>
> > I also found the KL divergence to be a bit unmotivated [...] It seems like the entire discussion about KL is pointless and it would be more straightforward to just use the behavior entropy definition.
>
> We agree with the reviewer that directly introducing the entropy improves the paper’s clarity and readability. We thank the reviewer for their assessment and updated a revised version of our manuscript that does not introduce the KL divergence. These changes are colored in magenta and can be found on page 3.
>
> ---
>
> > [...] the paper also does not ever make a claim that multimodal policies would be good [...]
>
> We thank the reviewer for their suggestion to enhance the motivation for policies capable of learning diverse behavior.  In response, we have incorporated additional points and references into the introduction of our paper. These changes are colored in magenta and can be found on page 1.
>
> Additionally, we summarized these points below:
>
> 1. **Improving Generalization:** If the learned policy overfits to a specific set of demonstrated behaviors, it may not generalize well to new situations. By exposing the model to diverse behaviors, the risk of overfitting is reduced, and the learned policy is more likely to capture the underlying principles of the task rather than memorizing specific trajectories [1, 2, 3].
> 2. **Enhancing Skill Transfer:** Learning diverse behaviors facilitates better skill transfer across different but related tasks. If the agent can imitate a wide range of behaviors, it is more likely to possess a set of skills that can be applied to various tasks, making it a more versatile and capable learner [1, 3].
> 3. **Unpredictability in competitive Games:** Predicting an opponent’s strategy in competitive games, such as table tennis, becomes much harder if the adversary has a diverse set of skills [4].
>
> ---
>
> > What is the argument for why we cannot or should not do something where we provide additional context to the policy on how we want the task to be performed?
>
> The primary argument against adopting goal-conditioned imitation learning (GCIL) [5, 6, 7] lies in the labor-intensive nature of context provision. Integrating contextual information often entails human annotation, a resource-intensive and challenging process to scale, particularly with extensive, unannotated datasets. We believe that the findings from our research contribute to mitigating this dependence on additional labeling efforts by highlighting methods that can learn diverse behaviors from datasets without human annotations.
>
>
> We would like to thank the reviewers again for assessing our work. We would be delighted to address any additional questions or concerns they may have.
>
> ---
>
> [1] Neural Probabilistic Motor Primitives for Humanoid Control, ICLR ‘19
>
> [2] InfoGAIL: Interpretable Imitation Learning from Visual Demonstrations, NeurIPS ‘17
>
> [3] One Solution is Not All You Need: Few-Shot Extrapolation via Structured MaxEnt RL, NeurIPS ‘20
>
> [4] Specializing Versatile Skill Libraries using Local Mixture of Experts, CoRL ‘21
>
> [5] Goal-conditioned Imitation Learning, NeurIPS ‘19
>
> [6] Goal-Conditioned Imitation Learning using Score-based Diffusion Policies, RSS ‘23
>
> [7] From Play to Policy: Conditional Behavior Generation from Uncurated Robot Data, ICLR ‘23

---

### Official Review · Reviewer_K6Tb · 2023-10-31

**Soundness:** 3 good
**Presentation:** 3 good
**Contribution:** 2 fair
**Rating:** 8
**Confidence:** 5

**Summary:**

As the interest in learning behaviors from natural human data raises, importance of imitation learning algorithms that can successfully learn from diverse and potentially multi-modal human behavior raises similarly. However, such algorithms are only recently gaining traction, and such there does not exist many benchmarks for properly comparing them. This situation is what D3IL aims to resolve, by creating a new benchmark to evaluate and compare imitation learning algorithms capable of learning multi-modal behavior.

The paper is divided the following sections: first the authors introduce the diversity metric used to evaluate the algorithms, which is an important component since the diverse, multi-modal behavior require a good notion of "coverage" of the behaviors. Then, they introduce the environments, baseline algorithms, and follow up by showing the performance of the algorithms and architectures on the tasks, both on terms of success rate and behavior diversity. Finally, they run a host of ablation experiments, such as limited data and impact of historical information.

**Strengths:**

This paper is a timely work since learning from diverse human data has shown major success in other fields such as natural language processing, and evaluation of candidate algorithm that can learn from diverse datasets is of vital importance at this moment. Here are the things this paper did well:

1. The benchmark is quite principled, and both the success metric and the diversity metric are well justified while being intuitive and implementable.
2. The list of baseline algorithms is also quite comprehensive, and covers the list of recent important developments in the space.
3. The set of ablation experiments run covers the primary points of interest, such as dataset size, visual/state based models, and impact of history.

Overall, this is a paper with a straightforward mission that achieves its goals well.

**Weaknesses:**

The paper, while quite strong on the execution, has some major shortcomings that can be improved in the future.
- A benchmark paper is useless without the environment codes and the data, which is absent from the supplementary materials. This is a major negative for this paper because we are being asked to judge it without being able to understand how easy it may be to run new algorithms against this benchmark.
- Another primary criticism is that all five environments are very simple tabletop environments, and thus the complexity of the algorithm needed to solve that may not be quite high. A better benchmark would involve multiple kinds of environments, involving 2D/3D environments, potentially with different intractable elements.
- One critical component missing from the evaluation is the required forward pass time or control frequency of the algorithms, which to my understanding is one of the largest disadvantages of diffusion-based models.

**Questions:**

What is the action space used by the environments? In the diffusion policy paper they show that diffusion policies are better for some absolute action spaces while being worse for others relative action spaces. Clarification as to that would be great.

---

> ### Author Response · Authors · 2023-11-16
> **Rebuttal by Authors**
>
> >This paper is a timely work since learning from diverse human data has shown major success in other fields […]
>
> We thank the reviewer for their positive feedback. We aim to thoroughly address the concerns and questions raised by the reviewers.
>
> ---
>
> >A benchmark paper is useless without the environment codes and the data […]
>
> We have included the codebase as a .zip file in the supplementary material. Furthermore, we have supplied a link to an anonymous GitHub repository, which can be accessed here: https://github.com/d3iltest/Temporary_D3IL
>
> ---
>
> >Another primary criticism is that all five environments are very simple tabletop environments, and thus the complexity of the algorithm needed to solve that may not be quite high. A better benchmark would involve multiple kinds of environments, involving 2D/3D environments, potentially with different intractable elements.
>
> We appreciate the reviewer's consideration and understand the concern about the simplicity of the tabletop environments. As Reviewer DMDV shared a similar concern, we kindly guide the reviewer to our reply to Reviewer DMVD. In that response, we addressed the concerns and pointed to Appendix D of the revised version of the paper which elaborates on the new tasks we plan to introduce in the camera-ready version.
>
> ---
>
> >One critical component missing from the evaluation is the required forward pass time or control frequency of the algorithms, which to my understanding is one of the largest disadvantages of diffusion-based models.
>
> We acknowledge the importance of providing forward pass/inference times to offer a more comprehensive understanding of the strengths and weaknesses of the different methods. Consequently, we have revised our manuscript and incorporated this information. You can find the results in Appendix B, highlighted in green.
>
> ---
>
> >What is the action space used by the environments? In the diffusion policy paper they show that diffusion policies are better for some absolute action spaces while being worse for others relative action spaces. Clarification as to that would be great.
>
> In our experiments, we opted for velocity control, representing relative action spaces, as opposed to position control (absolute action spaces). A revised version of our manuscript has been uploaded, encompassing a thorough comparison between absolute and relative action spaces. However, our findings differ from those presented in the diffusion policy paper, as we observed consistently better performances with relative action spaces. The cause for this discrepancy is currently unknown. These additional experiments can be found in Appendix B and are highlighted in green.
>
> We welcome the opportunity to address any additional concerns or questions that the reviewers may have.

---

### Author Response · Authors · 2023-11-22
**Rebuttal Summary**

We extend our sincere appreciation to all reviewers for dedicating their time and effort to assess our paper, and for their valuable suggestions that have significantly enhanced the quality of our work. Moreover, we are glad that all reviewers have voted in favor of acceptance. We would like to reiterate the primary concerns during the review and elaborate on how we have addressed each of them.

1. **Code Accessibility.** Reviewers K6Tb and DMVD suggested making our codebase open-source to facilitate result reproduction and enhance the usability of our work. In response, we have included the codebase as a .zip file in the supplementary material. Furthermore, we have supplied a link to an anonymous GitHub repository.

2. **Task Complexity.** Reviewers K6Tb and DMVD recommended adding more complex tasks to leave more room for future research endeavors. In response, we designed two additional tasks, collected data, and provided preliminary results that show that state-of-the-art methods are not able to solve the tasks reliably.

3. **Motivation.** Reviewer S4Qj suggested providing further motivation as to why diverse behaviors matter. We have incorporated additional points and references into the introduction of our paper.

---

### Meta-Review · Area_Chair_8o4h · 2023-12-06

**Metareview:**

(a) Summary
The paper focuses on addressing the challenge of imitation learning (IL) in robotics, particularly in replicating the inherent diversity in human behavior. The authors introduce a simulation benchmark and corresponding Datasets with Diverse human Demonstrations for Imitation Learning (D3IL). The key contributions include:
1. Designing environments that involve multiple sub-tasks and manipulation of multiple objects, thereby increasing behavioral diversity.
2. Introduction of tractable metrics to evaluate the ability of IL algorithms to capture and replicate diverse behaviors.
3. Comprehensive evaluation of state-of-the-art IL methods, shedding light on their effectiveness in capturing multi-modal human behaviors.

(b) Strengths:

(+) Benchmark Design (Reviewer K6Tb, S4Qj): The benchmark is well-principled, with intuitive and implementable metrics for success and diversity. The environments are detailed for reproducibility, and model selection criteria are well documented.

(+) Comprehensive Evaluation (Reviewer K6Tb): The paper includes a wide range of baseline algorithms and conducts extensive ablation studies, covering dataset size, model types, and impact of history, offering valuable insights.

(+) Relevance and Timeliness (Reviewer K6Tb): The work is timely, aligning with the growing interest in learning from diverse human data, and is crucial for evaluating candidate algorithms in this emerging field.

(c) Weaknesses
(-) Lack of Environmental Complexity (Reviewer K6Tb, DMVD): The environments are limited to simple tabletop tasks, lacking variety in terms of 2D/3D environments and more complex elements, which could limit the assessment of algorithms' capabilities.

(-) Enumerated Behaviors Limitation (Reviewer S4Qj): The requirement to pre-define behaviors for the behavior-level entropy measure limits the approach’s applicability in more dynamic, real-world scenarios.

**Justification For Why Not Higher Score:**

While the paper introduces an important and timely benchmark, it doesn't have a strong enough contribution to warrant an oral. Had the paper included more complex environments across a range of robots, it might have been a suitable candidate for oral.

**Justification For Why Not Lower Score:**

There is a real need for datasets in robot learning with diverse human demonstrations being available and good evaluation metrics. This paper addresses this need. The paper is timely, written thoughtfully and is poised to play an important role in future papers in robot learning.

---

### Decision · Program_Chairs · 2024-01-16

Accept (poster)